# Flower-like patterns in multi-species bacterial colonies

**Liyang Xiong[1,2], Yuansheng Cao[1], Robert Cooper[2], Wouter-Jan Rappel[1], Jeff Hasty[2,3,4,5], Lev Tsimring[2,3]\***

[1]Department of Physics, University of California, San Diego, La Jolla, United States; [2]BioCircuits Institute, University of California, San Diego, La Jolla, United States; [3]The San Diego Center for Systems Biology, San Diego, United States; [4]Molecular Biology Section, Division of Biological Sciences, University of California, San Diego, La Jolla, United States; [5]Department of Bioengineering, University of California, San Diego, La Jolla, United States

**Abstract** Diverse interactions among species within bacterial colonies lead to intricate spatiotemporal dynamics, which can affect their growth and survival. Here, we describe the emergence of complex structures in a colony grown from mixtures of motile and non-motile bacterial species on a soft agar surface. Time-lapse imaging shows that non-motile bacteria 'hitchhike' on the motile bacteria as the latter migrate outward. The non-motile bacteria accumulate at the boundary of the colony and trigger an instability that leaves behind striking flower-like patterns. The mechanism of the front instability governing this pattern formation is elucidated by a mathematical model for the frictional motion of the colony interface, with friction depending on the local concentration of the non-motile species. A more elaborate two-dimensional phase-field model that explicitly accounts for the interplay between growth, mechanical stress from the motile species, and friction provided by the non-motile species, fully reproduces the observed flower-like patterns.

**\*For correspondence:** ltsimring@ucsd.edu

## Introduction

Microbial communities inhabit every ecosystem on Earth, from soil to hydrothermal vents to plants to the human gut (*Moyer et al., 1995*; *Gill et al., 2006*; *Fierer and Jackson, 2006*). They often form dense biofilms, whose structures are shaped by biological, chemical, and physical factors (*Stoodley et al., 2002*; *Flemming et al., 2016*; *Stubbendieck et al., 2016*). In the wild, most biofilms are comprised of multiple bacterial strains. They feature a diverse repertoire of social interactions, including cooperation (*Ben-Jacob et al., 2000*; *Griffin et al., 2004*), competition (*Hibbing et al., 2010*), and predation (*Jürgens and Matz, 2002*). Bacteria often signal, sense, and respond to each other through secondary metabolites (*Traxler et al., 2013*) or antibiotic compounds (*Garbeva et al., 2014*), and co-cultures can even exhibit different motility from either species on its own (*McCully et al., 2019*). These interactions may lead to the emergence of complex spatial structures, which can have a profound effect on bacteria survival and function, and promote biodiversity by optimizing the division of labor within the biofilm (*Nadell et al., 2016*). Spatial structure can also enhance horizontal gene transfer among different species (*Cooper et al., 2017*).

In addition to biochemical interactions, mechanical forces also play an important role in shaping the structure of bacterial communities. In dense colonies, bacteria push against each other due to growth and motility. Bacteria can exploit these mechanical interactions to adapt to the environment. For example, mechanical stresses cause buckling in *Bacillus subtilis* biofilms that allows them to improve nutrient transport and consumption (*Asally et al., 2012*; *Trejo et al., 2013*; *Wilking et al., 2013*). Although the role of mechanical interactions in single-species colonies has been studied

**eLife digest** Communities of bacteria and other microbes live in every ecosystem on Earth, including in soil, in hydrothermal vents, on the surface of plants and in the human gut. They often attach to solid surfaces and form dense colonies called biofilms. Most biofilms found in nature are comprised of many different species of bacteria. How the bacteria interact shapes the internal structures of these communities.

Many previous studies have focused on the molecules that bacteria use to relate to each other, for example, some bacteria exchange nutrients or release toxins that are harmful to their neighbors. However, it is less clear how direct physical contacts between bacteria affect the whole community.

*Escherichia coli* is a rod-shaped bacterium that is a good swimmer, but has a hard time moving on solid surfaces. Therefore, when a droplet of liquid containing these bacteria is placed in a Petri dish containing a jelly-like substance called agar, the droplet barely expands over a 24-hour period. On the other hand, a droplet containing another rod-shaped bacterium known as *Acinetobacter baylyi* expands rapidly on agar because these bacteria are able to crawl using microscopic "legs" called pili.

Here, Xiong et al. set out to investigate how a colony containing both *E. coli* and *A. baylyi* developed on a solid surface. The experiments showed that when a droplet of liquid containing both species was placed on agar, both species grew and spread rapidly, as if the *E. coli* hitchhiked on the highly motile *A. baylyi* cells. Furthermore, the growing colony developed a complex flower-like shape. Xiong et al. developed mathematical models that took into account how quickly each species generally grows, their ability to move, the friction between cells and the agar, and other physical properties. The models predicted that the *E. coli* cells that accumulate at the expanding boundary of the colony make the boundary unstable, leading to the flower-like patterns.

Further analysis suggested that similar patterns may form in other situations when motile and non-motile species of bacteria are together. These findings may help us understand the origins of the complex structures observed in many naturally occurring communities of bacteria.

previously (*Volfson et al., 2008*; *Xavier et al., 2009*; *Kearns, 2010*; *Boyer et al., 2011*; *Persat et al., 2015*), dynamics of multi-species communities driven by mechanical forces have received much less attention. Since bacterial strains can have significant differences in their growth and motility characteristics, one can expect the development of highly-heterogeneous mechanical stress distribution, which in turn can result in a complex spatiotemporal dynamics of the colony.

To study the interactions between bacterial species with distinct biological and physical properties, we choose *Acinetobacter baylyi*, a gram-negative bacterium that easily moves on soft surfaces using twitching motility (*Harshey, 2003*; *Bitrian et al., 2013*; *Leong et al., 2017*), and an *Escherichia coli* strain that is almost non-motile on soft agar. Additionally, wild-type *A. baylyi* possesses a Type VI Secretion System (T6SS) that enables them to kill other bacteria (including *E. coli*) on direct contact (*Schwarz et al., 2010*; *Cooper et al., 2017*). We found that when these two strains are mixed together and inoculated on an agar surface, growing colonies develop intricate flower-like structures that are absent when either species is grown by itself.

To shed light on the mechanism behind this intricate pattern formation, we tested whether biological cell-cell communication or mechanical interaction between strains with different motilities played the key role. Experiments with *A. baylyi* mutants lacking T6SS showed that the pattern formation did not rely on this system. On the other hand, genetically impairing *A. baylyi* motility eliminated the patterns entirely. We also demonstrated that agar concentration affects cell motility and plays an important role in pattern formation. These findings suggested that the mechanical interactions between species are indeed primarily responsible for the pattern formation.

We then formulated and analyzed two models: a geometrical model of the colony boundary motion and a 2D phase-field model of the entire colony, to describe the mechanical interactions between two species. Our results show that growth and cell motility differences are sufficient to explain the emerging patterns. Since the mechanism of flower-like pattern formation is rather general, it may be broadly generalizable to other multi-species colonies.

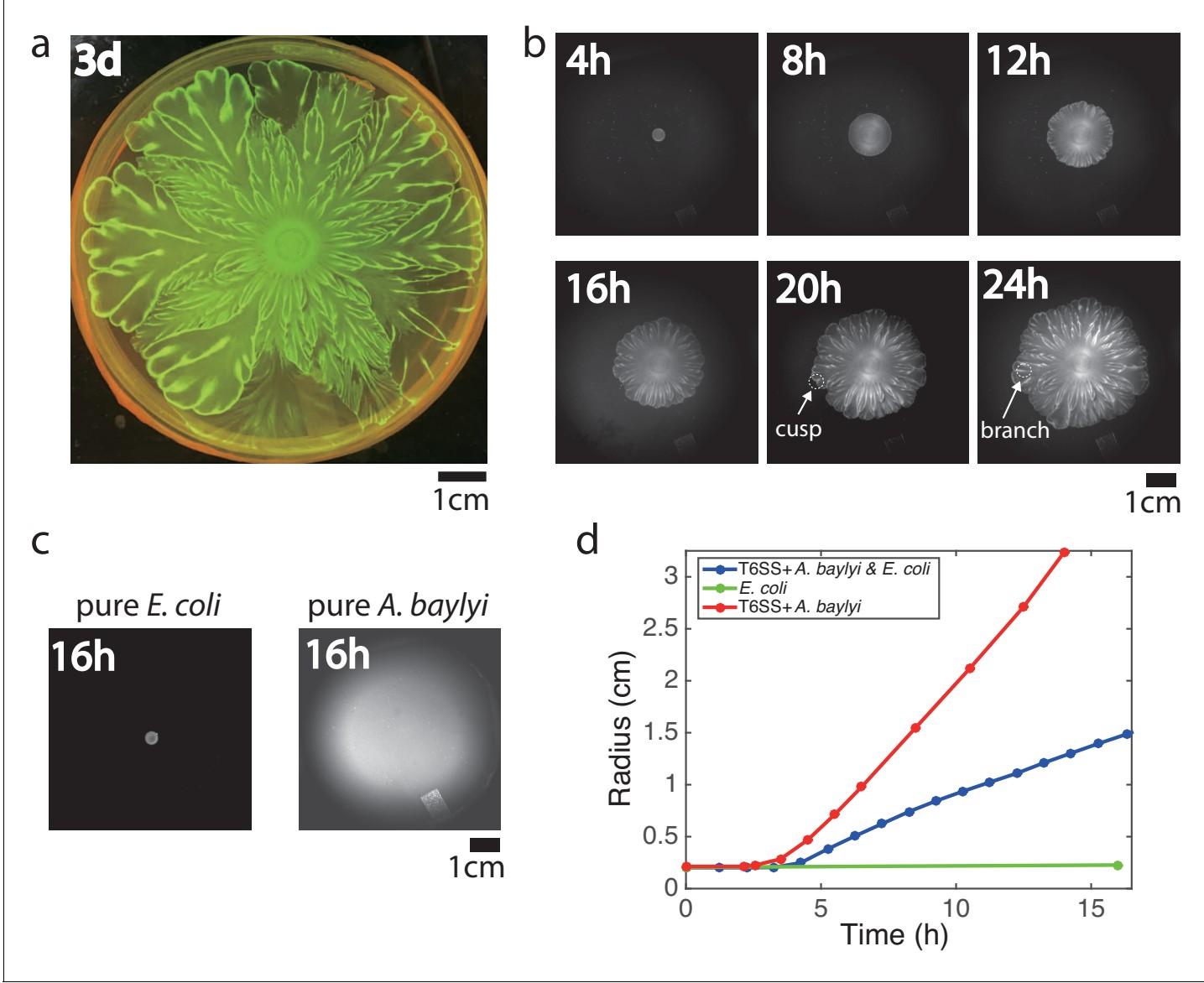

**Figure 1.** Flower-like patterns in mixtures of *E. coli* and *A. baylyi*. (a) The pattern after 3 days of growth on a 0.5% LB agar surface. (b) Time-lapse bright-field images of the developing pattern. (c) Pure *E. coli* and pure *A. baylyi* colonies show no patterns. (d) Radius of the colony vs time for pure *E. coli* (green), pure *A. baylyi* (red), and the mixture of *E. coli* and *A. baylyi* (blue). The radius is defined as $\sqrt{Area/\pi}$ where *Area* is the area of the colony which is calculated after image segmentation.

The online version of this article includes the following figure supplement(s) for figure 1:

**Figure supplement 1.** Bright-field image (left) and mTFP channel image (right) for the flower-like pattern after 24 hr of growth under milliscope.

**Figure supplement 2.** Time-lapse microscopic phase-contrast images after a mixture of *E. coli* and *A. baylyi* was inoculated on LB agar.

## Results

### Flower-like patterns in mixtures of *A. baylyi* and *E. coli* on nutrient-rich soft agar

We inoculated a mixture of *E. coli* and *A. baylyi* cells with an initial density ratio of 10:1 at the center of a Petri dish filled with soft LB agar (0.5% agar). To distinguish the two strains, we labeled *E. coli* with constitutively expressed mTFP. After growing at 37 °C for 3 days, this colony developed an intricate flower-like pattern (*Figure 1a*). To see how such patterns form, we tracked the colony growth with time-lapse imaging (*Figure 1b*, *Video 1*). Up to 8 hr after inoculation, the expanding colony

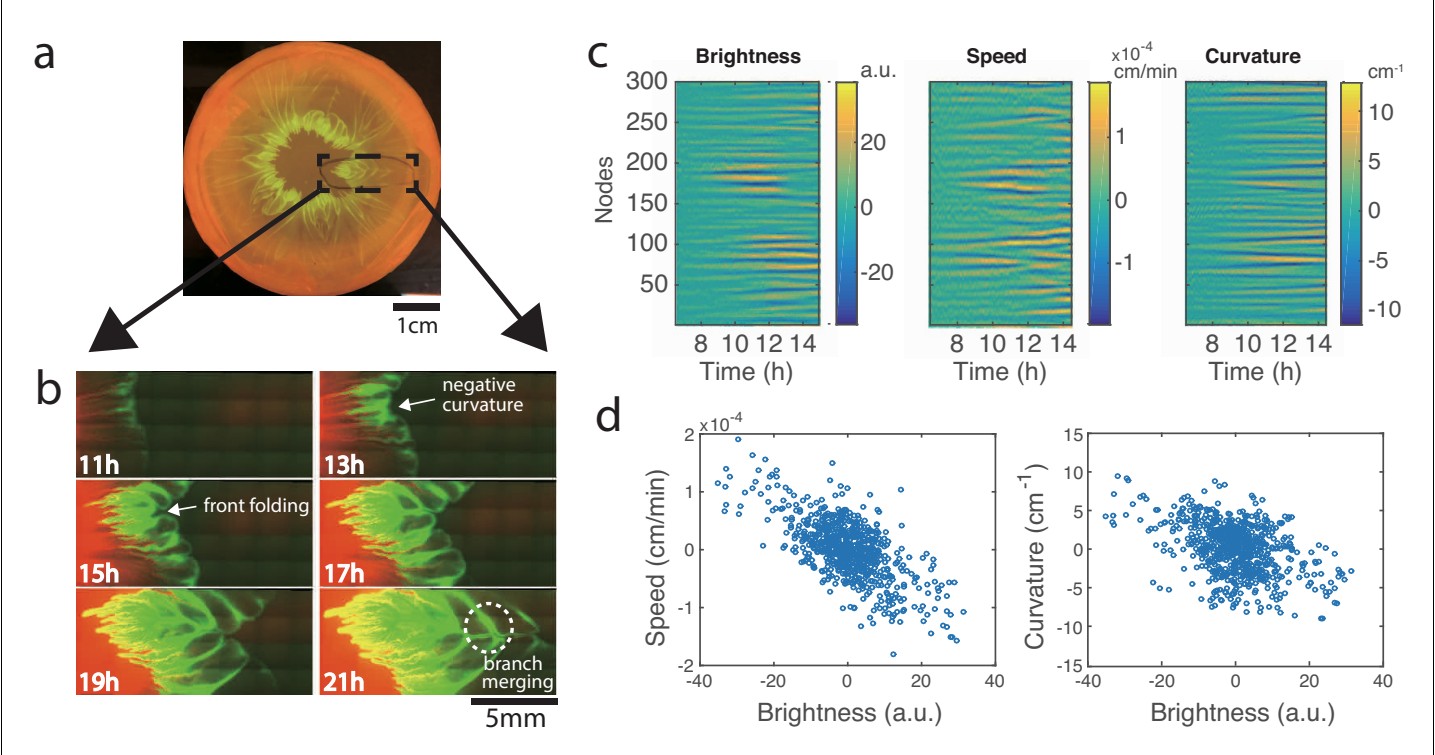

**Figure 2.** Development of branches in a growing pattern. (**a**) The whole colony in a Petri dish after one day. (**b**) Time-lapse microscopic images of the front propagation leading to branch formation and merging. (**c**) Kymographs of detrended brightness, front speed and front curvature along the colony boundary. (**d**) Scatter plots for detrended brightness vs speed (left) and detrended brightness vs curvature (right). Each circle corresponds to one virtual tracking node at one time point.

The online version of this article includes the following figure supplement(s) for figure 2:

**Figure supplement 1.** Pattern structure under mTFP and mCherry channels.

**Figure supplement 2.** Examples of the tracked colony boundary and traces of 300 virtual nodes on the colony boundary.

**Figure supplement 3.** An example of the detrended brightness, speed and local cuvature for all 300 nodes after 10 hr of colony growth in experiment.

remained nearly uniform and circular. Then the colony front began to visibly undulate. As the colony expanded further, the undulations grew and formed cusps that in turn would leave behind tracks (or 'branches'). These branches then merged, following the movement of cusps along the interface as the colony continued to expand. The branches were visible even in bright-field imaging, but they were also bright in the teal fluorescence channel, indicating that branches consisted of relatively more *E. coli* cells (*Figure 1—figure supplement 1*).

To test whether these flower-like patterns originate from interactions between the two species, we grew each species separately on the same 0.5% LB agar surface. The *E. coli* motility on agar is small, and the colony size remained relatively unchanged after 16 hr of growth (*Figure 1c*, left). After the same time, a colony of highly motile *A. baylyi* reached the edge of the plate (*Figure 1c*, right). In neither case did patterns emerge, showing that the flower-like pattern formation was a result of inter-species interaction. We measured the sizes of mixed, pure *E. coli* and pure *A. baylyi* colonies at different times after inoculation (*Figure 1d*). After an initial growth period in which cells filled the surface in a complete monolayer, the colony began to expand (an example is shown in *Figure 1—figure supplement 2*). The expansion speed of mixed colonies fell between those of pure *A. baylyi* and pure *E. coli* colonies, and the speed did not change much once the colonies began expanding.

## *E. coli* destabilize colony front by hindering *A. baylyi* expansion

To observe the pattern formation at higher resolution, we modified the experimental setup to fit under a fluorescence microscope (see Materials and methods). After 24 hr of growth, a droplet of 1:1 mixture of *E. coli* (expressing mTFP) and *A. baylyi* (expressing mCherry) grew into a clearly-visible

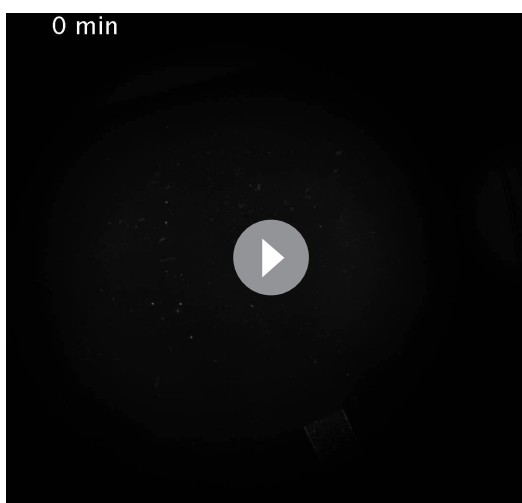

**Video 1.** Formation of flower-like patterns in the mixture of T6SS⁺ *A. baylyi* and *E. coli* under milliscope. Initial A:E density ratio was 1:10 and the cells grew on 10 mL LB agar (0.5% agar).
https://elifesciences.org/articles/48885#video1

**Video 2.** Development of branches in a growing flower-like pattern under microscope (4x magnification). Initial A:E density ratio was 1:1 and the cells grew on 10 mL LB agar (1% agar).
https://elifesciences.org/articles/48885#video2

flower-like pattern (*Figure 2a*). By zooming in on the front of the expanding colony, we were able to track the formation and merging of branches that gave rise to the flower-like structure of the patterns (*Figure 2b*, *Video 2*). While *A. baylyi* killed most *E. coli* via T6SS within the center of the inoculum, a significant number of *E. coli* managed to survive at the periphery where they were not in direct contact with *A. baylyi*. *E. coli* also has a higher growth rate ($1.53 \pm 0.11$ h$^{-1}$, $n = 3$) than *A. baylyi* ($1.13 \pm 0.01$ h$^{-1}$, $n = 3$), so by the time the colony began to expand, *E. coli* cells had already grown near the colony boundary which resulted in a band of *E. coli* around the expanding colony of mostly *A. baylyi* (*Figures 2b*, 11h).

As the colony kept expanding, in regions with more *E. coli* cells near the front, the expansion was slower, so the interface began to curve inward (*Figures 2b*, 13h). As the undulations grew bigger, the *E. coli* in the regions lagging behind became more concentrated, thus slowing down the local front advance even more. Eventually, the front folded onto itself near these stagnant regions and formed narrow 'branches' that continued to grow outward with the expanding colony front (*Figures 2b*, 15h, 17h). Later, the front with the branches folded again, and the previous branches merged inside the new fold (*Figures 2b*, 19h, 21h). Since *E. coli* continued to grow at the expanding colony front, new undulations and branches constantly appeared, and eventually a macroscopic, flower-like pattern of growing and converging branches formed. From *Figure 2—figure supplement 1*), it can be seen that the branches predominantly consisted of *E. coli* cells.

To quantify the effect of local *E. coli* concentration on the colony expansion, we analyzed the time-lapse images in *Figure 1b* (see Materials and methods). We adapted a boundary tracking program for eukaryotic cells (*Skoge et al., 2010*) to track the boundary of the bacterial colony. The colony boundary was parameterized by 300 virtual 'nodes' connected by springs (*Machacek and Danuser, 2006*). For each node, we measured local brightness (a proxy for *E. coli* concentration), front speed and front curvature. To offset the non-uniformity of the illumination and the overall change in speed and curvature for a growing colony, we detrended the data. The kymographs of these quantities for each node are shown in *Figure 2c*. Then we computed correlations between these quantities within the time window when the pattern began to form (about 9.5–11.5 hr after inoculation). As shown in *Figure 2d* (left), the brightness and expansion speed show strong anti-correlation (Pearson coefficient ρ=−0.67). This result confirms that higher *E. coli* density slows down the front propagation. Variations in the front speed lead to variations of the local curvature, and the scatter plot between brightness and curvature indeed shows significant anti-correlation (*Figure 2d* right, Pearson coefficient ρ=−0.43).

## Robustness of flower-like patterns to perturbations

First, we explored the effect of the initial *A. baylyi*:*E. coli* (A:E) density ratio on the resulting pattern. We varied the ratio of *A. baylyi* to *E. coli* in the inoculum while maintaining the same total density of bacteria. We found that when the starting ratios are low (A:E = 1:100 and 1:10), flower-like patterns emerged, while at high ratios (10:1 and 100:1) the *E. coli* were completely eliminated and no

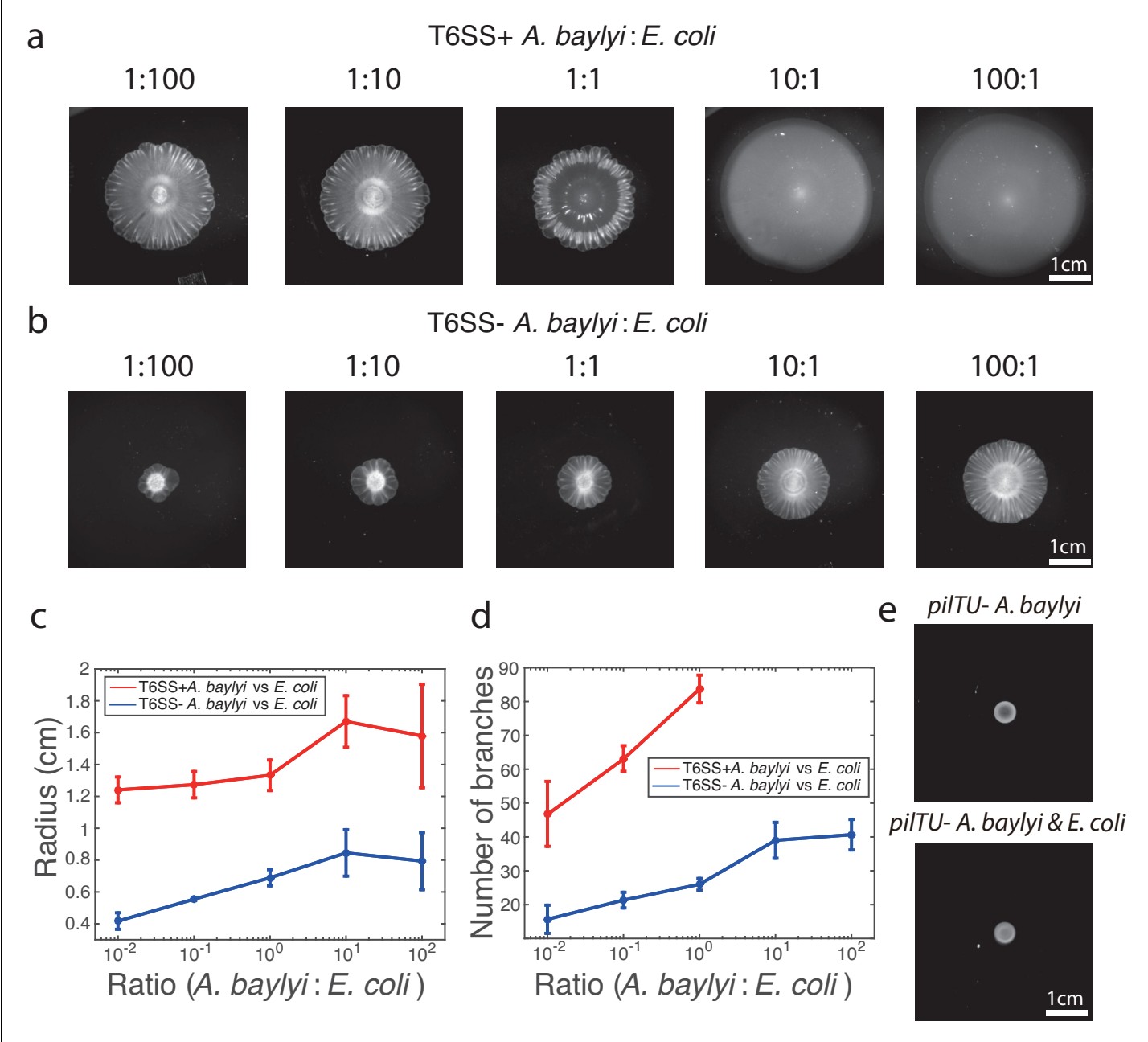

**Figure 3.** Pattern formation requires *A. baylyi* motility, but not killing. (**a-b**) Bright-field snapshots of colonies of T6SS⁺ a, and T6SS⁻ b, *A. baylyi* with *E. coli* 16 hr after inoculations at different initial density ratios. (**c**) The average colony radius vs density ratios 16 hr after inoculations. (**d**) Number of branches at the onset of front instability vs density ratios. (**e**) Colonies of pure *pilTU⁻* T6SS⁺ *A. baylyi* and the mixture of *pilTU⁻* T6SS⁺ *A. baylyi* and *E. coli* 16 hr after inoculation.

The online version of this article includes the following figure supplement(s) for figure 3:

**Figure supplement 1.** Colony radii after 16 hr of growth in 37 °C for pure T6SS+ A. baylyi, pure T6SS− A. baylyi, pure E. coli, mixture of T6SS+ A. baylyi and E. coli with 1:1 initial density ratio, mixture of T6SS− A. baylyi and E. coli with 1:1 initial density ratio with different agar concentrations (10 mL LB agar).

**Figure supplement 2.** Examples of the colonies for different combinations of *E. coli* and *A. baylyi* with different agar concentrations after 16 hr of growth on 10 mL LB agar.

**Figure supplement 3.** Microscope image of mixture of *E. coli* and T6SS⁻ *A. baylyi* on agar surface.

**Figure supplement 4.** Examples of the colonies for pure *pilTU⁻* T6SS⁺ *A. baylyi*, mixture of *pilTU⁻* T6SS⁺ *A. baylyi* and *E. coli* with initial seeding density ratio 1:1 with different agar concentrations after 16 hr of growth on 10 mL LB agar.

patterns formed (*Figure 3a*). At the intermediate ratio 1:1, *A. baylyi* dominated significantly at the center of the colony by killing *E. coli*, but the flower-like structure still developed at the colony periphery.

Second, we wondered whether T6SS-dependent killing played a role in the formation of these patterns when *E. coli* were not completely eliminated. We tested this by knocking out T6SS in *A. baylyi* (see Materials and methods for details). The growth rate of T6SS⁻ *A. baylyi* (1.09 ± 0.01 h⁻¹, n=3) was not significantly different from the wild type, but their motility was slightly lower as determined by colony expansion rate. Still, their motility remained much higher than *E. coli* (*Figure 3— figure supplement 1* and *Figure 3—figure supplement 2*). We inoculated mixtures of T6SS⁻ *A. baylyi* and *E. coli* with different initial ratios on 0.75% LB agar, and observed that the colony formed an outer ring of *E. coli* (*Figure 3—figure supplement 3*) and subsequently developed front instability, branches of *E. coli*, and a flower-like pattern in all cases (*Figure 3b*). The only qualitative difference between the T6SS⁻ and T6SS⁺ cases was that in the non-killing case, *E. coli* remained at a high concentration within the area of the initial inoculum. We measured the average radius of the colonies with different initial density ratios 16 hr after inoculations (*Figure 3c*, n = 3). In the case of a mixture of T6SS⁻ *A. baylyi* and *E. coli*, the more *E. coli* in the inoculum, the slower the colony expanded, which is consistent with our hypothesis that *E. coli* hinders the overall colony expansion. However, the trend is not as significant for the T6SS⁺ case, likely because T6SS⁺ *A. baylyi* kill most *E. coli* at the early stage, which increases and stabilizes the effective A:E ratio. We also counted the number of branches as they first emerged, when their circumferences were roughly the same, and found more branches in colonies seeded with less *E. coli* (*Figure 3d*, n = 3). In general, the overall structure of the patterns remained unchanged in the mixture of T6SS⁻ *A. baylyi* and *E. coli*. Thus, we concluded that the T6SS did not play a major role in the formation of flower-like patterns.

Third, the fact that two-species colonies expanded much more quickly than pure *E. coli* colonies strongly suggested that the high motility of *A. baylyi* is primarily responsible for the colony expansion. To test this hypothesis, we knocked out the *pilTU* locus of T6SS⁺ *A. baylyi*, which is required for the pilus-based twitching motility of *A. baylyi* (*Zhan et al., 2012*; *Leong et al., 2017*). As expected, colonies of *pilTU⁻ A. baylyi* cells did not expand significantly (*Figure 3e*, top) and did not form branching patterns when mixed with *E. coli* cells on 0.75% LB agar (*Figure 3e*, bottom). The results were the same when the colonies grew on other concentrations of LB agar (*Figure 3—figure supplement 4*). This demonstrates that the high *A. baylyi* motility plays a crucial role in the flower-like pattern formation.

Finally, we tested the pattern formation in mixtures of motile and non-motile *A. baylyi* (see Appendix 3). We found that flower-like patterns emerged in this case as well, which confirms the key role of the difference in motility for pattern formation. The patterns were less pronounced, but this can be probably explained by the fact that other physical parameters of non-motile *A. baylyi* (such as growth rates and effective friction) are more similar to motile *A. baylyi* than *E. coli*.

## Pattern-forming instability originates at the colony interface

Experiments showed that the formation of flower-like patterns appears to be preceded and caused by growing undulations of the colony front, where *E. coli* cells concentrate and locally slow expansion. To mechanistically understand how a ring of low-motility bacteria surrounding an expanding core of highly-motile bacteria can create such patterns, we turned to mathematical modeling. We adapted a one-dimensional 'geometrical' model of front dynamics (*Brower et al., 1983*; *Brower et al., 1984*) that casts the motion of the interface $\mathbf{x}(\sigma, t)$ in natural, reference-frame independent variables of curvature $\kappa$ and metric $g$ as a function of its arclength $s$ and time $t$ (see Appendix 1):

$$\dot{\kappa} = -\left(\frac{\partial^2}{\partial s^2} + \kappa^2\right)\mathcal{F}[\kappa, g], \quad \dot{g} = 2g\kappa\mathcal{F}[\kappa, g].$$

In the overdamped limit, the velocity functional $\mathcal{F} = (F_0 - F_s)/\mu(c)$ is determined by the balance of a constant outward force $F_0$ due to *A. baylyi* motility, surface tension $F_s = \gamma\kappa$ proportional to the interface curvature, and the resistance (friction) force $F_r = \mu(c)v$ that is proportional to the local velocity $v(s, t)$ with the friction coefficient $\mu(c)$ that in turn is proportional to the concentration of *E. coli* on the interface $c(s, t)$. Note that, in principle, nutrient depletion in the agar under the growing

colony and chemotaxis towards the developing nutrient gradient may also contribute to the outward force $F_0$, however it should not change the mechanism of the pattern-forming instability we are discussing here. All these forces are assumed to be normal to the interface (*Figure 4a*). For simplicity, in this interface model we ignore *E. coli* growth and leakage from the boundary into the interior and assume that the local interface concentration of *E. coli* is only changed by stretching or contraction of the interface, therefore $c$ should be inversely proportional to the square root of the metric $g$. A straightforward linear stability analysis demonstrates that the interface is indeed unstable to a broad spectrum of initial perturbations (for more details see Appendix 1).

To simulate the interface dynamics beyond the linear regime, we also constructed a discrete model of the continuous interface by replacing it with a closed chain of nodes connected by straight links (*Figure 4a* bottom). Each node carries a fixed amount of *E. coli*, so the local density of nodes per unit length of the interface corresponds to the local density of *E. coli*. Nodes are driven by a constant outwards expansion force $F_0$, surface tension, and a friction force that is proportional to the window-weighted average density of nodes per unit length. Additionally, we introduced short-range repulsive forces between nodes and between nodes and links, to prevent self-crossing of the interface. Detailed description of this model is also given in Appendix 1.

As an initial condition, we assumed that the chain forms a circle with nodes slightly perturbed from equidistant positions. *Figure 4b* shows time-lapse snapshots of the interface in a sample simulation (also see *Video 3*). *Figure 4c* shows the aggregate image of the interface during the colony expansion, with the color of a point corresponding to inverse local density of nodes when the interface passed through that point (also see *Video 4*). Assuming that a fixed fraction of *E. coli* is left behind the interface, this interface 'fossil record' should roughly correspond to the density of *E. coli* inside the colony. At the beginning, the interface remains nearly circular, but initial perturbations quickly grow as the colony expands, producing large front undulations. Regions with lower node density expand more quickly because they experience less friction, and this expansion stretches the chain and further reduces the node density per unit length, creating a positive feedback loop. Concave regions, on the contrary, accumulate nodes and thus move outward more slowly. Eventually, cusps are formed in these lagging regions that have very high node density and therefore move very slowly, if at all. The regions on both sides of the cusp continue to expand toward each other and

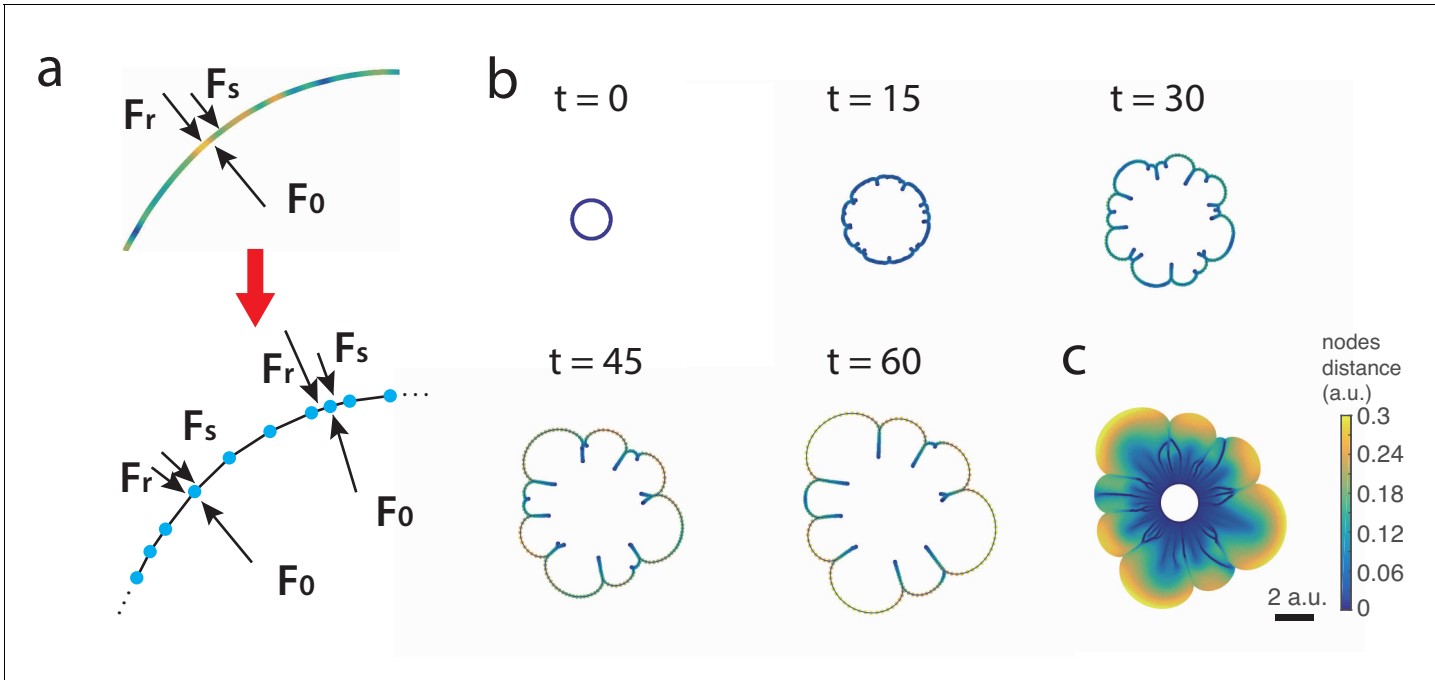

**Figure 4.** Discrete interface model. (a) Sketches of the continuum and discrete interface models. (b) Snapshots of the interface in discrete interface model for a sample simulation with parameters listed in Appendix 1. The colors of the nodes correspond to the distance between node and its neighbors. (c) 'Fossil record' of *E. coli* densitiy on the moving interface.

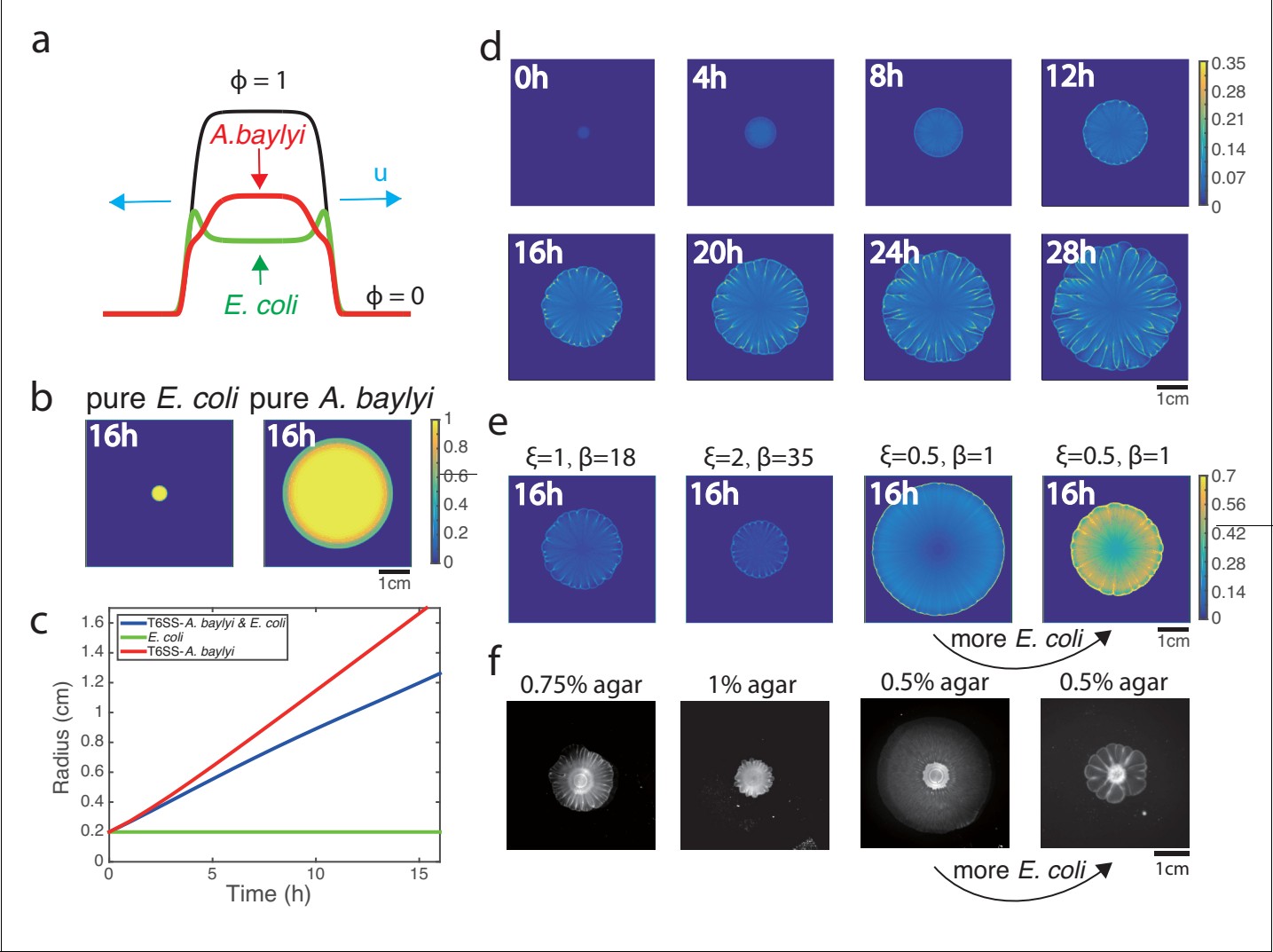

**Figure 5.** Phase-field model simulations of two-species colony growth. (a) Illustration of the model. (b) Snapshots of the colonies of pure *E. coli* and pure *A. baylyi* at *t* = 16. A colony of *E. coli* expanded only slightly, while a pure colony of *A. baylyi* expanded quickly, but remained circular. (c) Colony radius vs time for the mixed and single-species colonies. Radius is defined as $\sqrt{\text{colony area}/\pi}$. (d) Several snapshots of *E. coli* density during the growth of a mixed colony in simulations. (e) Colony snapshots at time *t* = 16 in simulations using different friction parameters. For larger friction, the colony grew slower, but still featured flower-like patterns. For smaller friction, the colony expanded more quickly, but patterns eventually disappeared. However, increasing the initial concentration of *E. coli* at low friction coefficients restored patterning. (f) Experimental snapshots with different agar concentrations 16 hr after inoculation: similar phenomenology observed.

The online version of this article includes the following figure supplement(s) for figure 5:

**Figure supplement 1.** Several snapshots of *A. baylyi* density during the growth of a mixed colony in a phase-field model simulation.

**Figure supplement 2.** Analysis of the colony boundary dynamics in phase-field model simulation.

eventually 'collide'. After collision they form 'double-layers' that remain nearly static and only increase in length as the overall interface expands further. Thus, 'branches' with high concentration of *E. coli* form. As the front continues to expand, the interface already containing branches continues to undulate and form new cusps. This causes the earlier branches to merge, similar to what we observed in experiments (*Figure 2*). These simulation results suggest that indeed branch formation and merging can be explained by mechanics of a resistive ring surrounding a colony, which is stretched by the colony expansion. However, since this model neglects *E. coli* growth, the average density of nodes per unit length gradually decays, and eventually, the front instability ceases, in divergence with experimental results. To account for cell growth as well as for the diffusive leakage

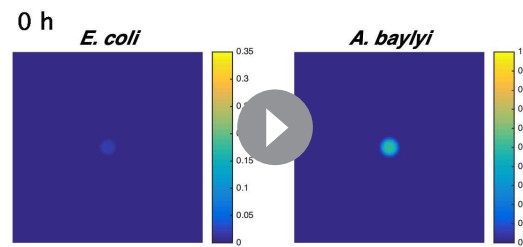

**Video 5.** A sample simulation of the phase-field model of two-species colony growth.
https://elifesciences.org/articles/48885#video5

of *E. coli* from the interface into the bulk of the expanding colony, we developed a more elaborate 2D model of the growing multi-species colony.

## Phase-field model of flower-like pattern formation

We also developed a more detailed two-dimensional, multi-component model of the expanding bacterial colony that is conceptually similar to the phase-field models used for description of eukaryotic cell motility and migration (*Shao et al., 2010*; *Shao et al., 2012*; *Camley et al., 2013*) (*Figure 5a*). It is based on PDEs for the densities of *A. baylyi* $\rho_A$ and *E. coli* $\rho_E$, together with an equation that describes the velocity field **u** of the colony. This velocity field drives the expansion of the colony and is generated by a combination of stress due to cell growth and motility, viscosity, and bottom friction that is dependent on local *E. coli* density. The resulting free boundary problem is solved using the phase-field method, which introduces another PDE for an auxiliary field $\phi$ that changes continuously from 1 inside the colony to 0 outside (see Appendix 2 for the detailed formulation of the model). The boundary is then automatically defined as $\phi = 1/2$ and can thus be computed without explicit tracking techniques.

When we initialized the model with small circular domains of either pure *E. coli* or *A. baylyi*, the colony boundaries remained circular, and no patterns emerged (*Figure 5b*). Consistent with the experiments, the *E. coli* colony only slightly expanded, while the *A. baylyi* colony expanded rapidly (*Figure 5c*). When we initialized the model with a mixture of *A. baylyi* and *E. coli*, the colony grew at an intermediate speed (*Figure 5c*), as in the experiments (*Figure 1d*). The mixed colony simulations also exhibited front instability leading to formation of branches of *E. coli* (*Figure 5d*, the snapshots of *A. baylyi* are shown in *Figure 5—figure supplement 1*, also see *Video 5*). As the colony grew, the branches merged and expanded, and a flower-like pattern developed. The *E. coli* density, colony boundary curvature and expansion speed can be analyzed using the same method we used for experimental data shown in *Figure 2c,d*, which also shows the anti-correlation between *E. coli* density and local speed (*Figure 5—figure supplement 2*).

Agar concentration is known to have a strong effect on the motility of bacteria (*Harshey, 2003*) and their adhesion to the agar surface (*Kolewe et al., 2015*), so we reasoned that in our phase-field model changing agar concentration could be simulated by changing friction parameters. The frictional force in our model consists of two contributions: a small basal friction (characterized by parameter $\xi$) and stronger contribution proportional to the local *E. coli* concentration with coefficient $\beta$. Thus, to mimic different agar concentrations, we varied both $\xi$ and $\beta$. The leftmost panel in *Figure 5e* shows the colony snapshot at $t = 16$ for the same parameter values as the time-lapse sequence in *Figure 5d*. The next panel corresponds to larger $\xi$ and $\beta$ (presumably, higher agar concentration), where as expected, the colony expanded slower. The third panel shows the snapshot for smaller $\xi$ and $\beta$ (lower agar concentration), in which case the colony expands fast, but no patterns emerge. However, for the same low $\xi$ and $\beta$, when we started a simulation from 10x higher *E. coli* density, the friction provided by *E. coli* increased, and patterning re-emerged (*Figure 5e*, fourth panel).

These numerical predictions were fully validated by experiments in which we varied the agar concentration and the initial density ratio of *E. coli* and T6SS⁻ *A. baylyi*. The leftmost panel in *Figure 5f* shows the snapshot of the colony started from 1:1 mixture after 16 hr of growth on 0.75% agar surface. When we increased the agar concentration to 1% (*Figure 5f*, second panel), the colony expanded slower but the flower-like pattern emerged. Conversely, for low agar concentration (0.5%), colony grew fast but patterns were completely eliminated (*Figure 5f*, third panel). However, for the same 0.5% agar concentration but A:E = 1:100 initial density ratio, the flower-like pattern formation was rescued (*Figure 5f*, fourth panel).

# Discussion

Motility plays a key role in the local spread of bacteria. In this paper, we studied the structure of growing colonies comprised of two bacterial species, *E. coli* and *A. baylyi*, with very different motilities. Not only did the highly-motile species (*A. baylyi*) accelerate the spread of the slow species (*E. coli*), but the structure of the expanding colony quickly became highly heterogeneous and eventually produced very intricate, flower-like patterns.

Bacterial colonies can expand on a surface in a variety of ways, assisted by volumetric pressure from cell growth and division, multiple types of motility (*Harshey, 2003*), chemotaxis (*Golding et al., 1998*; *Ben Amar, 2013*), osmotic pressure gradients from the extracellular matrix (*Seminara et al., 2012*; *Dilanji et al., 2014*; *Srinivasan et al., 2019*), secretion of surfactants that assist wetting (*Kearns, 2010*; *Trinschek et al., 2017*), etc, and these mechanisms are not mutually exclusive. In our case, we found that the key, necessary driver for expansion of mixed *A. baylyi/E. coli* colonies is the motility of *A. baylyi*. The expansion force appears to be mediated by cells physically bumping into and pushing each other, as colonies do not begin to expand outward until they reach a near confluent monolayer density (*Figure 1—figure supplement 2*). Before this point, motility in the interior can simply result in cell rearrangement, but once a confluent monolayer is reached, growth combined with motility begins to push the boundary outward. In our models, the effective expansion and friction forces are physically and experimentally motivated, but it is unclear to what extent the effective forces result from true friction, wetting forces, etc. In future work, it would be interesting to explore the detailed mechanistic underpinnings of these forces.

Pattern formation in growing colonies of single bacterial species has been studied extensively (*Fujikawa and Matsushita, 1989*; *Budrene and Berg, 1991*; *Golding et al., 1998*; *Matsushita et al., 1998*), and branching patterns were often found in these experiments. The emergence of these patterns is usually driven by nutrient limitation and ensuing chemotaxis, with agar concentration also having a strong effect on their morphology. For example, colonies expand homogeneously on soft agar rich with nutrients, but under nutrient limitation and in semi-solid agar, complex patterns emerge (*Budrene and Berg, 1991*; *Matsushita et al., 1998*; *Golding et al., 1998*). In our system, however, we used rich LB media, and single-species colonies in the same conditions did not produce patterns, suggesting that the mechanism of pattern formation here is different.

Cell killing via the T6SS is an important ecological interaction, but it did not appear to play a major role in the formation of these patterns. We found no significant differences in pattern formation with T6SS$^+$ and T6SS$^-$ strains of *A. baylyi*. In fact, we did not observe noticeable killing of *E. coli* by T6SS$^+$ *A. baylyi* after a short initial period (*Video 2*). We believe that an extracellular matrix

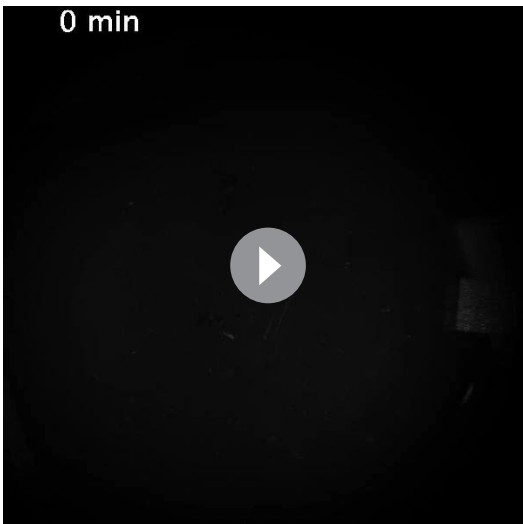

**Video 6.** When T6SS$^+$ *A. baylyi* and *E. coli* were inoculated separately on 10 mL LB agar (0.75% agar), the flower pattern formed only in a segment.
https://elifesciences.org/articles/48885#video6

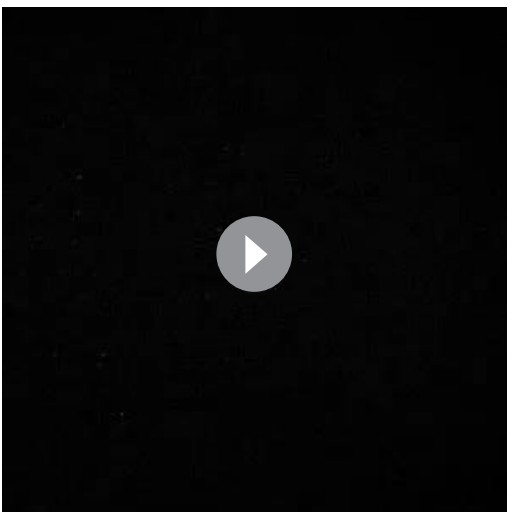

**Video 7.** Segmentation and tracking of the boundary of the growing colony from *Video 1*.
https://elifesciences.org/articles/48885#video7

may have played a role here, as recent studies showed that it protected bacteria from T6SS attacks from other species (*Toska et al., 2018*; *Molina-Santiago et al., 2018*). Overall, our experiments and modeling provided strong evidence in favor of the mechanical nature of the pattern-forming instability, arising from the interplay between outward pressure generated by the growth and high motility of *A. baylyi*, and the friction provided by sessile *E. coli* that adhere to the agar surface.

Ecologically, one of the primary challenges for any species is to maximize its geographic dispersal. Motility enables bacteria to escape from local stresses, move to locations with more nutrients, or invade host tissue (*Harshey, 2003*). However, motility, especially on hard surfaces, requires additional gene expression which could be a metabolic burden (*Kearns, 2010*). So some bacteria take advantage of other species with larger motility to colonize new niches. For example, by hitchhiking on zooplankton, water-borne bacteria can reach places that are otherwise inaccessible for them due to density gradients (*Grossart et al., 2010*). Non-motile staphylococcal species hitchhike on swimming bacteria such as *Pseudomonas aeruginosa* (*Samad et al., 2017*). Motile swarming *Paenibacillus vortex* was shown to transport non-motile *Xanthomonas perforans* (*Hagai et al., 2014*) or *E. coli* (*Finkelshtein et al., 2015*) on agar surfaces. In our system, *A. baylyi* cells move by twitching instead of swarming, and our results suggest that slow-moving bacteria might take advantage of fast-moving twitching species by hitchhiking, or 'surfing' along the expanding boundary, and thus spread farther. This can be seen clearly from the experiment in which *E. coli* and *A. baylyi* were inoculated separately at a small distance on agar surface (*Video 6*). The *A. baylyi* colony expanded and pushed *E. coli* to places where *E. coli* alone could not reach.

The flower-like patterns appear to require a combination of several factors: motility of one of the two species, hitchhiking of the non-motile species with the motile one, and sufficiently strong effective friction from the non-motile strain. Indeed, no patterns form without motility of one of the species, see *Figure 3e*. Hitchhiking appears to be a necessary but not sufficient condition for flower-like pattern formation. Indeed, without hitchhiking *E. coli* would simply be left behind and not present in the expanding colony of *A. baylyi*. On the other hand, we observed that *E. coli* also hitchhiked in round colonies (e.g. *Figure 5f*, third panel), where patterns did not form, presumably because *E. coli* did not exert sufficiently strong effective friction, due to properties of the agar or too low cell density. In the phase-field model, lowering *E. coli*-dependent friction corresponds to reducing parameter $\beta$, and indeed, for small $\beta$, patterns do not form. Additionally, higher growth rate of the non-motile strain facilitates formation of a dense ring around the expanding colony, which makes patterns more robust. We also observed flower-like patterns when *E. coli* was replaced with non-motile *A. baylyi* strain, but they were less robust, presumably because the growth rate of the non-motile *A. baylyi* strain was lower than that of the motile one, although it is also possible that the effective friction of non-motile *A. baylyi* could be less than that of *E. coli* (*Appendix 3—figure 1*).

Although *E. coli* and *A. baylyi* may not necessarily find themselves in the same ecological niche, bacteria with different motilities are ubiquitous in the environment (*Harshey, 2003*). Therefore, the mechanisms of codependent motility and pattern formation described here are likely to be broadly applicable in natural habitats or even have implications in the transmission of pathogenic microbes. For example, *Acinetobacter baumannii*, an increasing threat in hospitals due to multi-drug resistance (*Dijkshoorn et al., 2007*), is closely related to *A. baylyi* (*Touchon et al., 2014*), also has twitching motility (*Eijkelkamp et al., 2011*; *Clemmer et al., 2011*), and coexists with *E. coli* in at least one known niche, namely hospitals. Thus, the generic pattern-formation and hitchhiking described here may be quite common in diverse environments.

## Materials and methods

### Strains

We used *E. coli* MG1655 and *A. baylyi* ADP1 (ATCC #33305). The *E. coli* strain carried a plasmid that constitutively expressed mTFP and a kanamycin resistance gene. *A. baylyi* had a kanamycin resistance gene and the mCherry gene integrated in the genome. We also constructed a T6SS⁻ *A. baylyi* (Δhcp) mutant by first fusing the tetracycline resistance marker (TetA) from pTKS/CS to approximately 400 bp homology arms amplified from either side of hcp (ACIAD2689) in the *A. baylyi* genome, and mixing the donor oligo with naturally competent *A. baylyi*. The *pilTU*⁻ strain was

constructed similarly to delete the genes ACIAD0911-0912. All *A. baylyi* strains used in this study retain their endogenous immunity genes to T6SS attack.

## Culture conditions and image capturing

*E. coli* and *A. baylyi* cells were taken from −80 °C glycerol stocks, inoculated in LB with appropriate antibiotics (kanamycin for *E. coli* and T6SS$^+$ *A. baylyi*, tetracycline for T6SS$^-$ *A. baylyi*) and grown at 37 °C separately. When their OD600 reached about 0.3, both *E. coli* and *A. baylyi* were concentrated to OD = 1, still separately. They were then mixed at specified volume ratios, and 3 μL was inoculated on the surface of 10 mL LB agar in the center of an 8.5 cm Petri dish. The plate was incubated at 37 °C. The images were taken using a custom 'milliscope' fluorescence imaging device unless indicated otherwise.

When the colony development was to be observed under a microscope, a 5.5 cm Petri dish was used with 15 mL 1% base agar (without LB) and top 10 mL LB agar (1% agar). After the cell culture was inoculated and dried, it was put on the stage of an inverted, epifluorescence microscope (Nikon TI2). The magnification was 4X. Fluorescent images were acquired using a 4X objective and a Photometrics CoolSnap cooled CCD camera in a 37 °C chamber. The microscope and accessories were controlled using the Nikon Elements software.

The bacteria growth rates were measured in a Tecan plate reader.

## Colony tracking

We adapted the method and the MATLAB code from *Skoge et al. (2010)* to track the colony boundary. The bright-field images were first segmented to identify the colony using an active contour method (*Chan et al., 2000*). The segmentation result is illustrated in *Video 7*. Then the colony boundary pixels were interpolated by a closed cubic spline and the boundary was parameterized by 300 virtual nodes, which were evolved in time as a coupled spring system (*Figure 2—figure supplement 2*) (*Machacek and Danuser, 2006*). For each node, three quantities were measured: brightness, extension speed and curvature. Brightness at each node was defined as the median of the neighboring pixels assigned to each node (see *Skoge et al., 2010*). Extension speed was computed by the displacement of a node from *t* to *t*+50 min. Curvature was calculated by taking derivatives of the spline contour. Then the time series of these quantities were detrended as following: At each time point, fast Fourier transform (FFT) is carried out for each variable across all nodes and in the resulting transform, the first few low frequencies are set to zero. Then inverse FFT is carried out to obtain the detrended values for each variable at each node. After detrending, all variables can be negative at certain nodes. An example of these quantities for all nodes at a particular time point is shown in *Figure 2—figure supplement 3*. In *Figure 2d*, we sampled 7 time points with 20 min interval from 9.5 hr to 11.5 hr and for each time point we plotted 100 nodes.

## Mathematical models

Detailed description of the two models is given in Appendices 1 and 2.

## Acknowledgements

We thank Megan Dueck for building the custom 'milliscope' used in our study, Philip Bittihn for helpful discussions, and Kit Pogliano lab for providing the original *E. coli* strain. This work was supported by the National Institutes of Health (grant R01-GM069811), the National Science Foundation (grant PHY-1707637), San Diego Center for Systems Biology (NIH grant P50-GM085764) and the DOD Office of Naval Research (grant N00014-16-1-2093).

## Additional information

### Competing interests

Jeff Hasty: Jeff Hasty has competing financial interest in GenCirq, Inc. The other authors declare that no competing interests exist.

## Funding

| Funder | Grant reference number | Author |
| --- | --- | --- |
| National Institutes of Health | R01-GM069811 | Lev Tsimring<br>Liyang Xiong<br>Robert Cooper<br>Jeff Hasty |
| National Science Foundation | PHY-1707637 | Yuansheng Cao<br>Wouter-Jan Rappel |
| National Institutes of Health | San Diego Center for Systems Biology (P50-GM085764) | Lev Tsimring<br>Liyang Xiong<br>Robert Cooper<br>Jeff Hasty |
| Office of Naval Research | N00014-16-1-2093 | Lev Tsimring<br>Liyang Xiong |

The funders had no role in study design, data collection and interpretation, or the decision to submit the work for publication.

## Author contributions

Liyang Xiong, Conceptualization, Data curation, Software, Formal analysis, Validation, Investigation, Visualization, Methodology, Writing—original draft, Writing—review and editing; Yuansheng Cao, Robert Cooper, Software, Investigation, Methodology, Writing—original draft, Writing—review and editing; Wouter-Jan Rappel, Jeff Hasty, Resources, Supervision, Funding acquisition, Writing—review and editing; Lev Tsimring, Conceptualization, Resources, Software, Formal analysis, Supervision, Funding acquisition, Writing—original draft, Project administration, Writing—review and editing

## Author ORCIDs

Liyang Xiong (iD) https://orcid.org/0000-0002-3257-7643
Yuansheng Cao (iD) https://orcid.org/0000-0002-6857-6044
Robert Cooper (iD) https://orcid.org/0000-0003-2136-0403
Wouter-Jan Rappel (iD) https://orcid.org/0000-0003-3833-7197
Lev Tsimring (iD) https://orcid.org/0000-0003-0709-3548

## Decision letter and Author response

Decision letter https://doi.org/10.7554/eLife.48885.sa1
Author response https://doi.org/10.7554/eLife.48885.sa2

# Additional files

## Supplementary files

- Source code 1. Colony boundary tracking software.
- Transparent reporting form

## Data availability

All data generated or analysed during this study are included in the manuscript and supporting files.

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

## Appendix 1

# Interface model

## Continuous interface dynamics

To describe the motion of the interface separating the growing bacterial colony from the environment, we can use the framework originally proposed by *Brower et al. (1983)*; *Brower et al. (1984)* for solidification patterns. We assume that the motion of the interface is a result of the local balance of the 'pushing force' and the frictional force that is linearly proportional to the local interface velocity. The 1D interface (a closed line) at time $t$ is specified by the position vector $\mathbf{x}(t, \sigma)$ where $0 \leq \sigma \leq 1$ is the variable parametrizing the interface such that $\mathbf{x}(t, 0) = \mathbf{x}(t, 1)$. Using the 'orthogonal gauge' assumption that the velocity $d\mathbf{x}/dt$ is orthogonal to the tangent vector $\boldsymbol{\tau} = \partial\mathbf{x}/\partial\sigma$, the equation of motion for the interface in the overdamped limit can be written in the general form

$$\mu \frac{d\mathbf{x}}{dt} = \hat{\mathbf{n}} F(\mathbf{x}, \partial\mathbf{x}/\partial\sigma, ...) \tag{1}$$

where $\hat{\mathbf{n}}$ is the unit vector normal to the interface at $\mathbf{x}$ (perpendicular to $\boldsymbol{\tau}$), $F$ is the normal force that generally may depend on the overall interface position and other parameters, and $\mu$ is the friction coefficient. As *Brower et al. (1984)* demonstrated, this equation can be transformed to the reference-frame independent *local* equations of motion for the local curvature $\kappa$ and the curve metric $g = \boldsymbol{\tau} \cdot \boldsymbol{\tau}$ as a function of arclength $s$ and time $t$:

$$\dot{\kappa} = -\left(\frac{\partial^2}{\partial s^2} + \kappa^2\right)\mathcal{F} \tag{2}$$

$$\dot{g} = 2g\kappa\mathcal{F} \tag{3}$$

where $\mathcal{F} = F/\mu$, the arclength $s$ is given by

$$s = \int_0^\sigma \sqrt{g(\sigma')}d\sigma' \tag{4}$$

and the curvature is defined by

$$\kappa = -\hat{\mathbf{n}} \cdot \frac{\partial^2 \mathbf{x}}{\partial s^2} \tag{5}$$

Now we need to specify the driving force $F$ and the friction coefficient $\mu$ for our system in which a growing colony is surrounded by the thin band of highly frictional *E. coli* that hinders the colony expansion. We assume that $F$ depends only on the local curvature $\kappa$ in the following simple form:

$$F = F_0 - \gamma\kappa \tag{6}$$

This assumption will be violated if/when the interface will develop large folds and will attempt to 'collide' with each other, then non-local terms in $F$ become essential. We confine our continuous description here to sufficiently early times before this non-local interaction occurs. We postulate that the friction coefficient is a linear function of the local concentration of *E. coli*, $c$,

$$\mu = 1 + \alpha c \tag{7}$$

where without loss of generality we take $\mu = 1$ at $c = 0$. Under the additional simplifying assumption that the total amount of *E. coli* on the interface is conserved and neglecting their diffusion along the interface, the local concentration of *E. coli* will be inversely proportional to the square root of metric $g$, $c = c_0/\sqrt{g}$. In reality, of course, *E. coli* also grows and is left behind

in the bulk of the colony, but we ingore these effects in this simple model (see the phase-field model below where these effects are taken into consideration). Thus, the closed-form model for the interface expansion has the following form

$$\dot{\kappa} = -\left(\frac{\partial^2}{\partial s^2} + \kappa^2\right)\left(\frac{F_0 - \gamma\kappa}{1 + \alpha c_0/\sqrt{g}}\right) \tag{8}$$

$$\dot{g} = 2g\kappa\left(\frac{F_0 - \gamma\kappa}{1 + \alpha c_0/\sqrt{g}}\right) \tag{9}$$

We can perform a linear stability analysis of a flat interface ($\kappa = 0, g = 1$) by substituting ansatz

$$\kappa = Ke^{iks+\lambda t} \tag{10}$$

$$g = 1 + Ge^{iks+\lambda t} \tag{11}$$

in *Equations (8), (9)*. The Jacobian of the linearized system reads

$$J = \begin{bmatrix} -\frac{\gamma k^2}{1+\alpha c_0} & \frac{\alpha c_0 F_0 k^2}{2(1+\alpha c_0)^2} \\ \frac{2F_0}{1+\alpha c_0} & 0 \end{bmatrix}. \tag{12}$$

For positive $\gamma, \alpha$, one of the two eigenvalues of this Jacobian is always positive. At small wavenumbers $k$, it increases linearly with $k$,

$$\lambda = \sqrt{\frac{\alpha c_0}{(1+\alpha c_0)^3}}F_0 k \tag{13}$$

and for large $k$ it reaches the maximum value

$$\lambda_m = \frac{\alpha c_0 F_0^2}{(1+\alpha c_0)^2 \gamma} \tag{14}$$

Since the growth rate is positive for all values of $k$, this instability may lead to singularities in curvature (cusps). This is indeed what is found in numerical simulations of the discrete analog of this model (see the next section). These singularities correspond to the origins of 'branches' of *E. coli* that the interface leaves behind during the flower pattern growth.

## Flexible-chain interface model

The interface dynamics beyond linear instability stage can be analyzed numerically. Unfortunately, it is difficult to implement self-avoidance of the interface in the framework of the continuum model described in the previous section. Thus, we implemented a discrete flexible-chain model that is analogous to the continuum model described above but contains additional interaction terms between the nodes that prevent self-intersection of the chain. Specifically, we represent the interface as a closed chain of $N$ nodes with coordinates $\mathbf{x}_i, i = 1, ..., N$. Let us introduce the vectors connecting node $i - 1$ to node $i$ (we assume that node 0 is the same as node $N$): $\mathbf{\Delta}_i = \mathbf{x}_i - \mathbf{x}_{i-1}$. Each node is driven by the 'expansion force' $F_0$ that acts along the unit vector $\hat{\mathbf{n}}_i$ that is directed outwards along the bisectrix of two adjacent edges, $\mathbf{\Delta}_i$ and $\mathbf{\Delta}_{i+1}$. It is counteracted by the 'friction' force that is directed along $-\hat{\mathbf{n}}_i$ and is proportional to the local density of *E. coli* $c_i$ associated with node $i$ and by the surface tension force that is proportional to the local curvature of the interface $\kappa_i$. In addition, we introduce repulsion forces between all nodes and edges that prevent the interface from self-intersecting. The equation of motion in the overdamped limit can be written as follows:

$$\frac{d\mathbf{x}_i}{dt} = \hat{\mathbf{n}}_i \frac{F_0 - \gamma\kappa_i}{1 + \alpha c_i} + \sum_{j \neq i} \mathbf{f}_{ij}^{nn} + \sum_{j \neq i} \mathbf{f}_{ij}^{ne} \tag{15}$$

The discrete analog of the local curvature at node $i$ is defined as follows,

$$\kappa_i = \left| \frac{\mathbf{\Delta}_{i+1}}{\Delta_{i+1}} - \frac{\mathbf{\Delta}_i}{\Delta_i} \right| \frac{2}{\Delta_i + \Delta_{i+1}} \tag{16}$$

where $\Delta_i = |\mathbf{\Delta}_i|$.

We assume that each node carries the fixed 'amount' of *E. coli* $c$, and the local concentration of *E. coli* $c_i$ is defined as the average amount of $c$ per unit length of the interface. In the simplest case, it can be computed as $c/L_i$ where $L_i$ is the half-sum of lengths of two edges attached to node $i$, $L_i = (\Delta_i + \Delta_{i+1})/2$, however in simulations we typically used longer averaging over two adjacent edges on both sides,

$$c_i = \frac{2(2K+1)c}{\sum_{j=-K}^{K}[\Delta_{i+j} + \Delta_{i+1+j}]} \tag{17}$$

with $K = 2$.

The last two terms in the r.h.s. of **Equation (15)** represents the vector sum of possible repulsive forces acting on the node $i$ from other nodes ($\mathbf{f}_{ij}^{nn}$) or edges ($\mathbf{f}_{ij}^{ne}$) of the chain. The node-node force acts along the vector connecting nodes $i$ and $j$, $\mathbf{x}_i - \mathbf{x}_j$. We assume that the node-edge force acts perpendicular to the orientation of the $j$-th link, $\mathbf{\Delta}_j$. We assume that the node-node force $\mathbf{f}_{ij}^{nn}$ is zero if $d_{ij}^{nn} = |\mathbf{x}_i - \mathbf{x}_j| > d_0$ and varies as $F_m(1 - d_{ij}^{nn}/d_0)^4$ for $d_{ij}^{nn} < d_0$ with small $F_m \ll F_0$. Similarly, the node-edge force $\mathbf{f}_{ij}^{ne}$ is zero if the distance between the node $i$ and the edge $j$, $d_{ij}^{ne} > d_0$ and varies as $F_m(1 - d_{ij}^{ne}/d_0)^4$ for $d_{ij}^{ne} < d_0$.

## Parameters

We used parameters below (**Appendix 1—table 1**) unless specified otherwise.

**Appendix 1—table 1.** Parameters of the interface model.

| $F_0$ | $\alpha$ | $\gamma$ | $F_m$ | $d_0$ | $c$ | $N$ | $dt$ |
|---|---|---|---|---|---|---|---|
| 1 | 0.5 | $10^{-8}$ | 0.1 | 0.01 | 1 | 512 | 0.001 |

## Appendix 2

### Phase-field model

### Model description

In this more elaborate 2D model of a two-strain colony, we consider it as a growing mass of compressible two-component fluid. A convenient way to describe a compact expanding colony is to use a phase-field approach where the phase $\phi$ changes smoothly from 0 outside the colony to 1 inside. The evolution of phase field $\phi$ is described in earlier work (**Shao et al., 2012**). $\phi$ is given by the equation:

$$\frac{\partial \phi}{\partial t} = -\mathbf{u} \cdot \nabla \phi + \Gamma(\epsilon \nabla^2 \phi - G'(\phi)/\epsilon + \kappa \epsilon |\nabla \phi|) \tag{18}$$

where $\mathbf{u}$ is the velocity field, $\Gamma$ is a Lagrange multiplier, $\kappa = -\nabla \cdot (\nabla \phi / |\nabla \phi|)$ is the local interface curvature, and $\epsilon$ characterizes the interface width. The first term on the right-hand side is the advection term. The second one is the surface energy. In the third term $G(\phi) = 18\phi^2(1-\phi)^2$ is included to force the bistable dynamics of $\phi$ field with two stable fixed points at 0 and 1. The last term is added to cancel the surface energy and stablize the phase-field interface, as detailed in **Biben and Misbah (2003)** and **Biben et al. (2005)**. Note that in the interface model, we include the surface tension term $\gamma \kappa$ to stabilize the system, otherwise **Equation (13)** holds for all $k$ and $\lambda$ goes to infinity when $k$ increases.

Close inspection of the growing colony showed that the velocities of the two strains in close proximity are very similar, since the mixture of *E. coli*, *A. baylyi* and the (presumable) extracellular matrix is dense, liquid-like, and miscible. Therefore, we use a single local velocity, which represents the actual velocity of the bacterial cells, to describe the movement of two species.

The dynamics of the *A. baylyi* cells density $\rho_A$ within the colony is described by

$$\frac{\partial(\phi \rho_A)}{\partial t} + \nabla \cdot (\phi \rho_A \mathbf{u}) = \nabla \cdot (\phi D_A \nabla \rho_A) + \alpha_A \phi \rho_A (1 - \rho_A - \rho_E) \tag{19}$$

The second term in the left-hand side is the advection term while the two terms in the right-hand side are diffusion and growth terms respectively. $D_A$ and $\alpha_A$ are the diffusion constant and growth rate of *A. baylyi* respectively. The growth term follows logistic form and we assume that the growth can be saturated when the total density of *A. baylyi* ($\rho_A$) and *E. coli* ($\rho_E$) reaches 1. Note that the densities of two species are already scaled here.

Similarly, the dynamics for *E. coli* cells density $\rho_E$ is described by

$$\frac{\partial(\phi \rho_E)}{\partial t} + \nabla \cdot (\phi \rho_E \mathbf{u}) = \nabla \cdot (\phi D_E \nabla \rho_E) + \alpha_E \phi \rho_E (1 - \rho_A - \rho_E) \tag{20}$$

where $D_E$ and $\alpha_E$ are the diffusion rate and growth rate of *E. coli*. Note that the advection of the phase field and both cell densities is provided by the same velocity field $\mathbf{u}$.

The system is treated as a viscous Newtonian fluid (**Rubinstein et al., 2009**; **Shao et al., 2012**). The velocity field can be determined by the overdamped Stokes equation:

$$\nabla \cdot [\nu(\phi)(\nabla \mathbf{u} + \nabla \mathbf{u^T})] + \nabla \cdot (\chi \sigma_A) - [\xi + \beta f(\rho_E)\phi]\mathbf{u} = 0 \tag{21}$$

where $\nu(\phi) = \nu_0 \phi$ is the viscosity, $\sigma_A = -\eta \phi \rho_A \mathbf{I}$ is the stress provided by motile *A. baylyi* cells ($\mathbf{I}$ is the identity matrix). $\chi$ is a random number uniformly distributed between $1 \pm \Delta$, which adds noise to the stress driven by *A. baylyi*. Because pure *E. coli* colony expands very slowly and pure *A. baylyi* colony expands fast, we assume that the stress provided by *E. coli* is negligible compared to *A. baylyi*. Our experiments with mixtures of *E. coli* and *A. baylyi* show that regions where there are more *E. coli* move outward more slowly, so we assume that *E. coli* cells provide friction to prevent colony from expanding fast. This is described by the last term

in which $\xi$ is the basal friction constant and $f(\rho_E) = \rho_E$ determines how the friction is modulated by *E. coli* cells. Here we assume it is simply proportional to $\rho_E$.

In reality, as the colony expands, the nutrients in the media are expected to get depleted over time at the center of the colony. However, in the experiment, where we use rich LB media, interesting dynamics mainly happen at the colony boundary, and the pattern inside the colony does not change once it forms. Therefore, we do not include the nutrient diffusion and uptake in our model.

## Parameters

Parameters of simulations on 0.75% LB agar are shown in **Appendix 2—table 1**. Some of these parameters (such as growth rates $\alpha_E$ and $\alpha_A$) are known from experiments, while others had to be plausibly hypothesized. For example, the diffusion constants for bacterial motion are only known very roughly (**Budrene and Berg, 1991**; **Kim, 1996**), but since *A. baylyi* is motile and *E. coli* is not, we chose the diffusion constant of *A. baylyi* to be two orders of magnitude higher than that of *E. coli*.

**Appendix 2—table 1.** Parameters of the phase-field model.

| $\Gamma$ | $\epsilon$ | $D_A$ | | | $\alpha_A$ | $D_E$ | $\alpha_E$ |
|---|---|---|---|---|---|---|---|
| 0.008 cm/h | 0.16 cm | 0.0024 cm²/h | | | 1.2 hr⁻¹ | 4×10⁻⁵ cm²/h | 1.3 hr⁻¹ |
| $\nu_0$ | $\eta$ | $\xi$ | $\beta$ | $\Delta$ | $\Delta x$ | $\Delta y$ | $\Delta t$ |
| 0.0036 cm² | 0.03 cm²/h | 1 | 18 | 0.3 | 0.01 cm | 0.01 cm | 1×10⁻⁴ hr |

Note that if parameters $\nu_0, \eta, \xi, \beta$ are multiplied by the same constant factor, the velocity as determined by **Equation (21)** will not change. So we set arbitrarily $\xi = 1$ and chose other parameters $\nu_0, \eta, \beta$ relative to $\xi$. Based on the presence of sharp kinks in the developing front structure, we concluded that viscosity plays a minor role in the dynamics, so we chose the viscosity coefficient to be small. The value of $\beta$ is chosen based on fitting the average expansion rates of colonies of *A. baylyi* and *E. coli* mixtures.

Our simulations showed that diffusion and viscosity terms did not play significant roles in the dynamics. Changing $D_A$ had little effect on the colony expansion speed and the pattern formation (**Appendix 2—figure 1**). The reduction of $\nu_0$ makes colony expand faster but the flower-like pattern still forms (**Appendix 2—figure 1**). On the contrary, the stress and friction terms play major roles in our model. For the stress term, $\eta$ is chosen to make the expansion speed of pure *A. baylyi* colony similar to experimental measurement. We also added white uniformly-distributed noise (with magnitude $\Delta$) to the stress term to break the circular symmetry and induce the front instability. When $\Delta$ is small, the colony front instability also occurs, but at a later time point and merging of branches is not obvious (**Appendix 2—figure 1**, first row), so we choose $\Delta = 0.3$ in our simulations.

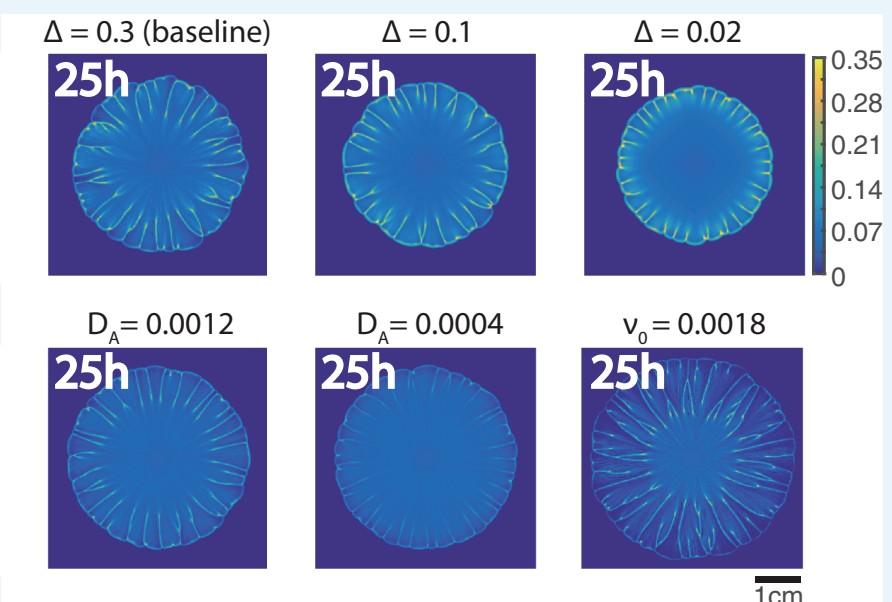

**Appendix 2—figure 1.** The influence of different parameters on the pattern formation in phase-field model. The parameters in *Appendix 2—table 1* are used for the baseline simulation. For each snapshot, only one parameter (the parameter on top of each snapshot) is changed relative to the baseline simulation while other parameters stay the same.

To model changes in the agar concentration (*Figure 5e*), we varied $\xi$ and $\beta$ while keeping $\nu_0$ and $\eta$ the same. As shown in *Figure 5e*, for the simulation in 0.5% LB agar, $\xi = 0.5, \beta = 1$ and for the simulation in 1% LB agar, $\xi = 2, \beta = 35$. The colony radii after 14 hr in simulations are illustrated in *Appendix 2—figure 2* which can be compared to *Figure 3—figure supplement 1*. Note that in *Figure 3—figure supplement 1*, we show the experimental data after 16 hr of growth because in experiments, the colonies only begin to expand 2 to 3 hr after inoculation, while in simulations the colonies begin to expand immediately.

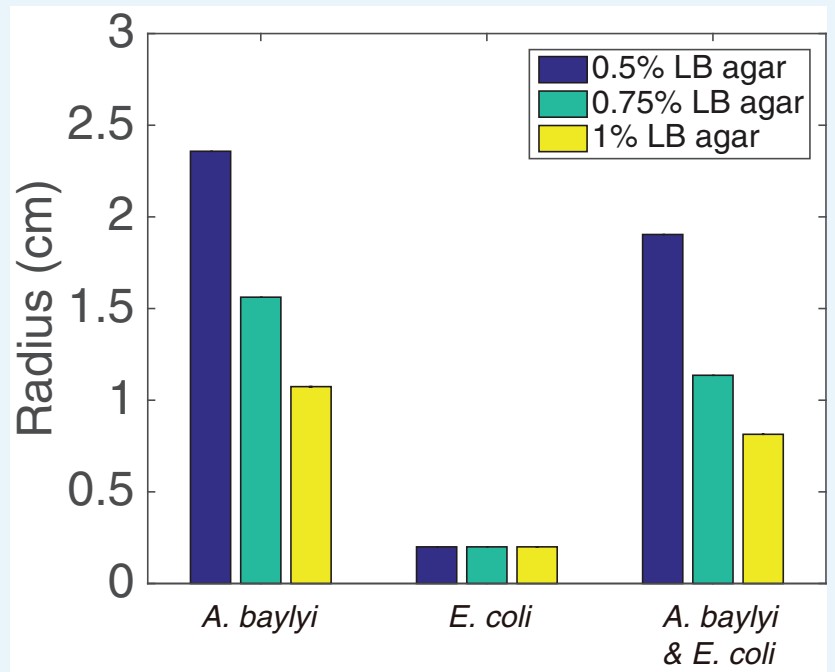

**Appendix 2—figure 2.** Colony radii after 14 hr of growth in simulations. The parameters in

*Appendix 2—table 1* are used for the 0.75% LB agar simulation. For 0.5% LB agar, $\xi = 0.5, \beta = 1$ and for 1% LB agar, $\xi = 2, \beta = 35$.

## Numerical algorithm

The numerical algorithm is similar to *Camley et al. (2013)*. For the initial conditions, we set $\phi = 0.5 + 0.5 \tanh[3(r_0 - r)/\epsilon]$ where $r_0 = 0.2$ cm and $r$ is the distance from the center of the simulation domain, so that $\phi$ is 1 inside and 0 outside of the colony. Initial $\rho_A$ and $\rho_E$ are proportional to $\phi$. We use periodic boundary conditions in the simulations.

We aim to solve *Equations (18)-(21)* with uniform spatial grid sizes $\Delta x, \Delta y$ and fixed time step $\Delta t$ from initial conditions $\phi^0, \rho_A^0, \rho_E^0, \mathbf{u}^0$. The system variables at time $t = n\Delta t$ are denoted as $\phi^n, \rho_A^n, \rho_E^n, \mathbf{u}^n$.

We first solve *Equation (18)* by forward Euler scheme:

$$\phi^{n+1} = \phi^n - \Delta t \mathbf{u}^n \cdot \nabla\phi^n + \Delta t \Gamma [\epsilon \nabla^2 \phi^n - G'(\phi^n)/\epsilon + \epsilon \kappa^n |\nabla\phi^n|]$$

with $\kappa^n$ calculated by $\kappa^n = -\nabla \cdot (\nabla\phi^n / |\nabla\phi^n|)$ when $|\nabla\phi^n| > 0.05$, and set to 0 otherwise.

The reaction-diffusion-advection equations for $\rho_A$ and $\rho_E$ are discretized using the forward Euler scheme:

$$\phi^n \frac{\rho^{n+1} - \rho^n}{\Delta t} + \frac{\phi^{n+1} - \phi^n}{\Delta t}\rho^n = \text{Advection} + \text{Diffusion} + \text{Reaction} \tag{22}$$

where $\phi^{n+1}$ is obtained from the above step, and $\rho^{n+1}$ is only updated when $\phi^n > 10^{-4}$. The advection term is calculated by

$$[\nabla \cdot (\phi^n \rho^n \mathbf{u}^n)]_{ij} = (\phi_{i+1/2,j}^n \rho_{i+1/2,j}^n u_{i+1/2,j}^n - \phi_{i-1/2,j}^n \rho_{i-1/2,j}^n u_{i-1/2,j}^n)/\Delta x$$
$$+ (\phi_{i,j+1/2}^n \rho_{i,j+1/2}^n v_{i,j+1/2}^n - \phi_{i,j-1/2}^n \rho_{i,j-1/2}^n v_{i,j-1/2}^n)/\Delta y$$

and for the diffusion term

$$[\nabla \cdot (\phi^n D \nabla \rho^n)]_{ij} = D[\phi_{i+1/2,j}\frac{\rho_{i+1,j} - \rho_{i,j}}{\Delta x} - \phi_{i-1/2,j}\frac{\rho_{i,j} - \rho_{i-1,j}}{\Delta x}]/\Delta x$$
$$+ D[\phi_{i,j+1/2}\frac{\rho_{i,j+1} - \rho_{i,j}}{\Delta y} - \phi_{i,j-1/2}\frac{\rho_{i,j} - \rho_{i,j-1}}{\Delta y}]/\Delta y$$

where $\mathbf{u} = (u, v)$, $\phi_{i\pm1/2,j} = (\phi_{i\pm1,j} + \phi_{i,j})/2$, $\phi_{i,j\pm1/2} = (\phi_{i,j\pm1} + \phi_{i,j})/2$, and we used the same definitions for $\rho$, $u$ and $v$ between collocation points. Then we can calculate $\rho^{n+1}$ from *Equation (22)*.

The Stokes equation *Equation (21)* is integrated by the semi-implicit Fourier spectral method (*Chen and Shen, 1998*; *Camley et al., 2013*) (to stabilize the scheme, we subtract the term $\nu_0\phi_0\nabla^2\mathbf{u}$ from both sides of Stokes equation with large constant $\phi_0$, e.g. $\phi_0 = 200$):

$$\xi\mathbf{u} - \nu_0\phi_0\nabla^2\mathbf{u} = \nu_0\nabla \cdot [\phi\nabla\mathbf{u}^T + (\phi - \phi_0)\nabla\mathbf{u}] + \nabla \cdot (\chi\sigma_A) - \beta f(\rho_E)\phi\mathbf{u}$$

To obtain $\mathbf{u}^{n+1}$, we set $\mathbf{u}_0^{n+1} = \mathbf{u}^n$ and solve the equation below iteratively using spectral Fourier method:

$$\xi\mathbf{u}_{k+1}^{n+1} - \nu_0\phi_0\nabla^2\mathbf{u}_{k+1}^{n+1} = \nu_0\nabla \cdot [\phi^{n+1}\nabla\mathbf{u}_k^{T,n+1} + (\phi^{n+1} - \phi_0)\nabla\mathbf{u}_k^{n+1}] + \nabla \cdot (\chi\sigma_A)^{n+1} - \beta f(\rho_E^{n+1})\phi^{n+1}\mathbf{u}_k^{n+1}$$

where $k = 0, 1, 2, \cdots$ are iteration steps. In simulations, we constrain the error by iterating the above process until

$$\max|\mathbf{u}_k^{n+1} - \mathbf{u}_{k-1}^{n+1}| < 0.01 \max|\mathbf{u}_k^{n+1}|$$

or until $k_{max} = 200$, and the final $\mathbf{u}^{n+1} = \mathbf{u}_m^{n+1}$.

## Appendix 3

# Mixtures of motile and non-motile *A. baylyi*

For an additional test of our hypothesis that the difference in motility between the two strains is indeed the key factor of the pattern formation, we mixed motile *A. baylyi* (T6SS⁻ or T6SS⁺) with non-motile (*pilTU⁻*) mutant of *A. baylyi* and inoculated them on 0.75% LB agar. Note that all *A. baylyi* strains used in this study have their endogenous T6SS immunity genes intact, so they do not kill each other (see Materials and methods).

In these experiments we also observed complex flower-like structures (see ***Appendix 3— figure 1*** and ***Appendix 3—video 1***), however the pattern formation was less robust than in the case of *A. baylyi/E. coli* mixtures. In particular, patterns were observed in the narrower range of initial density ratios for mixtures of motile T6SS⁻ *A. baylyi* and *pilTU⁻ A. baylyi*: Well-developed flower-like patterns were observed for initial density ratio $R$ = 1:10 (motile:non-motile), however, unlike the case of T6SS⁻ *A. baylyi* and *E. coli* mixtures, no patterns were observed for $R$ = 1:1, and only weak patterning was observed for $R$ = 1:100 (see ***Appendix 3—figure 1***, panels a-c). We hypothesize that the main reason for these differences is that in this case the non-motile strain did not have a faster growth rate. Non-motile *A. baylyi* has a significantly smaller growth rate ($1.03 \pm 0.12h^{-1}$, n = 3) than our *E. coli* strain ($1.53 \pm 0.11h^{-1}$, n = 3). The non-motile *A. baylyi* growth rate was actually even smaller than the growth rate of our motile strain, which may have been due to metabolic burden from the highly expressed *tetA* gene used to select them. Thus, for large $R$, motile *A. baylyi* 'outruns' the non-motile strain, which does not grow fast enough to first form a non-motile band around the colony. It is also possible that non-motile *A. baylyi* provide less friction (less adhesion to the agar surface) and that this also contributes to the differences in pattern formation with the case of *A. baylyi/E. coli* mixtures. This hypothesis is confirmed by the simulations of phase-field model (***Appendix 3—figure 2***). When the non-motile strain growth rate and non-motile strain-dependent friction are large, the pattern occurs. When one of the two parameters decreases, the patterns still persist while the pattern disappears if both growth rate and friction drop.

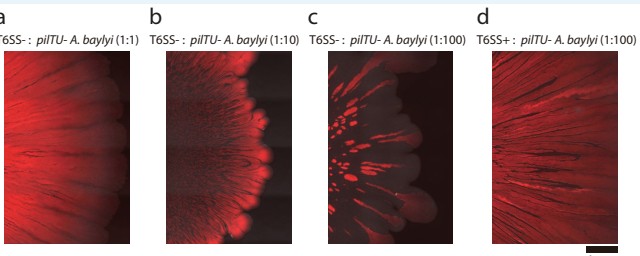

a
T6SS- : *pilTU- A. baylyi* (1:1)

b
T6SS- : *pilTU- A. baylyi* (1:10)

c
T6SS- : *pilTU- A. baylyi* (1:100)

d
T6SS+ : *pilTU- A. baylyi* (1:100)

1mm

**Appendix 3—figure 1.** Colonies of mixtures of motile (T6SS⁺ or T6SS⁻) and non-motile (*pilTU⁻*) *A. baylyi* after 18 hr of growth on 0.75% LB agar for different initial compositions, as indicated by the titles above the panels. Red color indicates fluorescent motile *A. baylyi* and dark regions within the colony indicate non-motile *A. baylyi* lacking fluorescent marker.

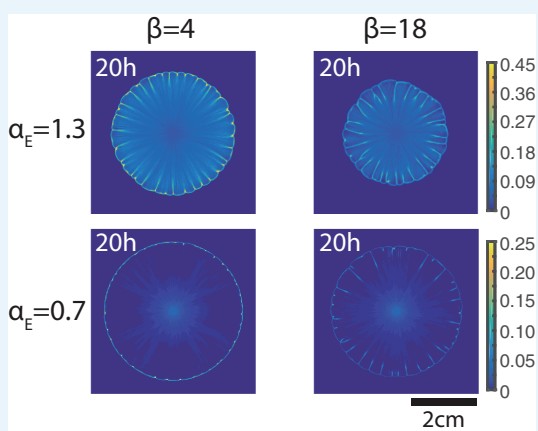

**Appendix 3—figure 2.** Non-motile strain density at time t = 20 in simulations of mixtures of motile and non-motile strains using different non-motile strain growth rate $\alpha_E$ and non-motile strain-dependent friction coefficient $\beta$.

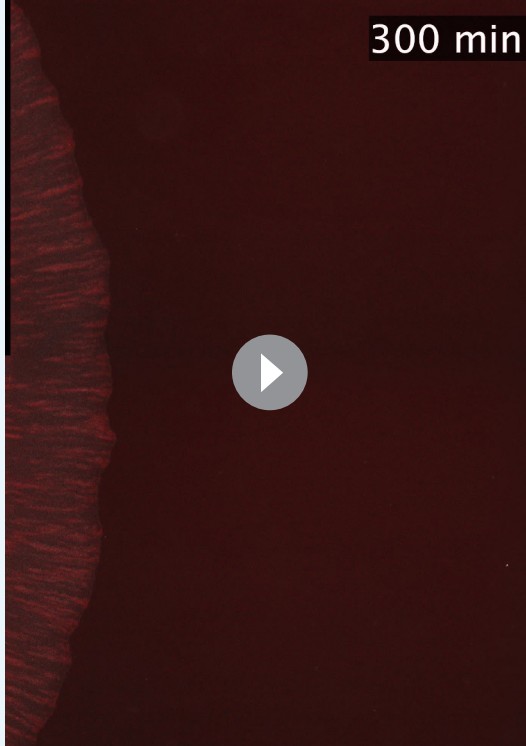

**Appendix 3—video 1.** Pattern formation in a mixture of motile T6SS⁻ and non-motile *pilTU⁻ A. baylyi* with intitial density ratio 1:10 on 0.75% LB agar.

We also found significant differences in patterning between mixtures of non-motile *A. baylyi* with T6SS⁺ or T6SS⁻ motile *A. baylyi*. When motile T6SS⁻ *A. baylyi* and non-motile *A. baylyi* are mixed with initial density ratio 1:100 (**Appendix 3—figure 1**, panel c), the non-motile strain dominates the colony and only weak patterns are observed, which are different from the earlier flower-like structures. However, when T6SS⁺ motile *A. baylyi* and non-motile *A. baylyi* are mixed, even with initial density ratio 1:100, T6SS⁺ motile *A. baylyi* dominate the colony (**Appendix 3—figure 1**, panel d). In this case, streaks of the non-motile strain (similar to those in flower-like patterns) can be observed, but they do not merge as in the earlier flower-like patterns. We believe that these differences are caused by the fact that our T6SS⁺ motile

*A. baylyi* has larger growth rate and motility than T6SS⁻, likely due to metabolic burden from the selection marker.

