## [Decision Letter]

**Acceptance summary:**

The paper describes the morphology of a two-species bacterial colony on soft agar, composed of twitching *Acinetobacter bayli* and non-motile *Escherichia coli*. When co-cultured, these colonies exhibit a flower-like pattern that is absent in pure cultures. The authors find that the type IV secretion system of *A. baylyi* is not essential for the observed morphology, whereas agar concentration strongly affects the pattern. Using continuous models of colony morphology they conclude differences in cell motility and growth are sufficient to explain the observed pattern.

**Decision letter after peer review:**

Thank you for submitting your article "Flower-like patterns in multi-species bacterial colonies" for consideration by *eLife*. Your article has been reviewed by three peer reviewers, and the evaluation has been overseen by a Reviewing Editor and Detlef Weigel as the Senior Editor. The following individuals involved in review of your submission have agreed to reveal their identity: Agnese Seminara (Reviewer #2).

The reviewers have discussed the reviews with one another and the Reviewing Editor has drafted this decision to help you prepare a revised submission.

The paper describes the morphology of a two-species bacterial colony on soft agar, composed of twitching Acinetobacter bayli and non-motile *Escherichia coli*. When co-cultured, these colonies exhibit a flower-like pattern that is absent in pure cultures. The authors find that the type IV secretion system of *A. baylyi* is not essential for the observed morphology, whereas agar concentration strongly affects the pattern. Using continuous models of colony morphology they conclude differences in cell motility and growth are sufficient to explain the observed pattern.

Overall, all the reviewers where very positive about the work. However, they raise a number of both experimental (mix motile/non-motile *A. baylyi*) and theoretical questions. Addressing all of these will greatly help the readers. I am attaching the reviews in full; please respond to all the points. As you see many of the theoretical points are similar between reviewers.

*Reviewer #1:*

In this manuscript, Xiong et al. describe how complex pattern formation arises within a 2-species bacterial biofilm. Specifically, they show that combining *E. coli* and *A. baylyi* results in flower-like patterns while neither species individually produces this effect. They use genetic perturbations to show that this effect requires motility (mixed biofilms with non-motile *A. baylyi* do not produce patterns) and does not depend on T6SS. They go on to reproduce the patterns using a mathematical model that captures the differences in motility between the 2 strains.

Overall, I thought this was a very nice and clearly presented study. The phenotype is quite interesting and the authors do a good job of providing a plausible explanation (mechanical effect resulting from differences in motility). I have only one experimental suggestion to confirm this explanation. Can the authors mix motile and non-motile *A. baylyi* and produce the patterns in a single species *A. baylyi* biofilm? If the floral patterns are really a result of motility differences (and not other unknown interspecies effects) than this should be possible. They have already created the non-motile *A. baylyi* mutant so it should be straightforward to try this experiment. If they observe this result, then I am enthusiastic for publishing the manuscript.

Reviewer #2:

I find the results interesting and convincing; I have a series of comments that I would like the authors to address before I can recommend this manuscript for publication. I am mainly concerned with justifying the choices made for the physical modeling and provide ideas to corroborate hitchhiking as a relevant evolutionary determinant. Below I detail my concerns, and suggest ways to improve clarity and strength of the results.

The experiments describing the evolution of the flower like pattern are clear and convincing. *E. coli* accumulates at boundaries and causes the colony to bend inward and fold onto this decelerating region creating cusps and branches. The speed of the boundary is anti-correlated with *E. coli* concentration and to curvature, corroborating the picture.

Flower like patterns are recovered by a simple model tracking the shape of the leading edge, assuming that each portion of the interface expands by a balance of friction, surface tension and an expanding force due to motility. Friction is proportional to concentration of (non-motile) E Coli, which causes instability.

1) It would be helpful to discuss the origin of the expansion force, since only part of the colony is motile. Are twitching cells pushing the other strain through cell-to-cell contact? (Can the authors provide a high-resolution image to show that cells are highly packed?) Or – given that an extracellular matrix is mentioned – is it entropically driven? (Is an extracellular matrix knock-out available to the authors?)

2) Are wetting forces negligible? An order-of-magnitude estimate of the different forces or experimental evidence would justify choice of these and not other forces.

3) In the more refined model, 2D concentrations of the two strains and of the phase field are evolved in time. I would imagine the two strains move relative to one another (*E. coli* remains attached to the substrate whereas *A baylyi* twitches). A two-fluid mixture appears a natural choice; the authors consider instead a single velocity field carrying all cells, and I am unclear what this velocity represents. Is there a way to relate it to actual velocities measurable experimentally, e.g. a weighted average of velocities of the two strains? Because I am confused about the meaning of v, I do not have an intuition why it follows Stokes equation.

4) The authors predict that at low friction, the colony expands quicker with no flower like pattern, unless a large concentration of *E. coli* is inoculated. Assuming agar concentration affects friction, these predictions are verified experimentally. A discussion is missing about the choice of ξ and β, which appears to me quite arbitrary. Also: is the value of η obtained by fitting the expansion rate a plausible value for a bacterial colony?

5) Hitchhiking is an appealing evolutionary advantage. The discussion would benefit from two additional points. First: the authors imply that flower-like patterns are directly related to hitchhiking. However from Figure 3A it looks like *E. coli* hitchhikes even in round colonies maybe less so than in flower-like colonies? Could you quantify this aspect? Second: what are the parameters that switch hitchhiking on/off in the model? Are there evidences that these parameters vary in different strains? Could genes control these parameters? This would help elucidate whether hitchhiking may be actively controlled (e.g. strains turning off motility would benefit from correspondingly turning on hitchhiking)

Reviewer #3:

The authors studied interactions between a motile and a non-motile bacterial species in growing colonies, and observed the development of complex patterns that were not present in colonies consisting only of one of these species. These patterns included an undulated interface of the colony that correlated with branched structures inside of the colony. The authors provided several lines of evidence that the speed of colony expansion correlates negatively with the local concentration of the non-motile bacteria. Based on these findings, they showed that a quite simple model describing the dynamics of the 1D colony interface can account for the formation of undulations and branches. This model essentially describes the advection and dilution of the non-motile species with the motion of the front of the motile bacteria, where a local friction increases with the concentration of non-motile bacteria. To additionally account for growth and diffusion of the bacteria, the authors further introduce a more complex 2D phase field model. Similar to the 1D interface model, patterns similar to the experimentally observed ones could be qualitatively reproduced.

While I am not an expert on biofilms, I found this manuscript interesting and well written. I found particularly intriguing that pattern formation based on the collective motion of two different bacteria species can be at least qualitatively accounted for by a simple 1D model.

In the following, I will focus on discussing the mechanical modeling part, which falls more in my area of expertise. While the interface model has appeal due to its simplicity, there are a few things that need to be better clarified if not corrected about it (see below). The phase field modeling seems to be mostly appropriate with respect to the assumptions made, but also here at least some clarifications would be good, mostly related to the field Φ (see below).

More detailed comments:

1) A brief discussion on possibilities of where the force F_0_ and the "active pressure" in Equation 22 could originate would be good (also in the initial phase where the colony is not expanding (Figure 1D)). In subsection “Pattern-forming instability originates at the colony interface” the authors suggest some kind of active pressure created by the *A. Baylyi* motility. However, it is not clear to me how this alone could explain an initial phase where the colony is not expanding at all (Figure 1D). The latter seems to be more consistent with a picture of nutrient consumption combined with chemotaxis (as proposed in earlier work, Discussion section paragraph two), which could also effectively create such an F_0_.

2) There are problems or at least a lack of clarity related to the way the friction force F_r_ is included in both 1D models describing the interface behavior.

a) The authors assume F_r_ to be proportional to the concentration of *E. coli*, c, but independent of velocity. As a consequence, if the concentration is high enough, the surface would move inward (even without surface tension F_s_=0), driven by this friction force. How realistic is such a friction force?

b) The normal velocity F[κ, g] is computed directly as a difference of normal forces (Equation 8). Apparently, the authors have implicitly assumed some additional, velocity-dependent friction here (and set the friction coefficient to one). It would be good to comment on how this additional friction is motivated.

c) As an alternative, one could assume F_r_ itself to depend linearly on velocity, a common assumption for motile cells on a substrate, which is also used in the authors' phase-field model (Equation 22). In this case and without additional friction, through force balance (F_0_-F_s_-F_r_=0) the normal velocity F[κ, g] would be given by a quotient between F_0_-F_s_ and a friction coefficient that is a function of c.

3) Even though the phase field approach is similar to previous work, some explanations, in particular related to the field Φ would make the manuscript more self-contained.

a) In subsection “Phase-field model of flower-like pattern formation” the authors state that Φ is introduced to avoid computational difficulties of dealing with the boundary. However, these difficulties and hence the necessity of Φ is nowhere explained in more detail. Why are ρ_A_ and ρ_E_ not sufficient?

b) Explanations on Equation 19 can be expanded on. In particular, brief comments on the second, diffusion-like term and the last term on the right-hand side could be helpful for readers.

c) The last term in Equation 19 is known to cancel the surface tension effect created by the second term (e.g. Biben and Misbah, 2003). Why do the authors explicitly remove surface tension in their phase field model while it is present in their interface models?

---

## [Author Response]

Reviewer #1:In this manuscript, Xiong et al. describe how complex pattern formation arises within a 2-species bacterial biofilm. Specifically, they show that combining *E. coli* and *A. baylyi* results in flower-like patterns while neither species individually produces this effect. They use genetic perturbations to show that this effect requires motility (mixed biofilms with non-motile A. baylyi do not produce patterns) and does not depend on T6SS. They go on to reproduce the patterns using a mathematical model that captures the differences in motility between the 2 strains.Overall, I thought this was a very nice and clearly presented study. The phenotype is quite interesting and the authors do a good job of providing a plausible explanation (mechanical effect resulting from differences in motility). I have only one experimental suggestion to confirm this explanation. Can the authors mix motile and non-motile A. baylyi and produce the patterns in a single species A. baylyi biofilm? If the floral patterns are really a result of motility differences (and not other unknown interspecies effects) than this should be possible. They have already created the non-motile A. baylyi mutant so it should be straightforward to try this experiment. If they observe this result, then I am enthusiastic for publishing the manuscript.

Thank you for a very interesting suggestion that we followed. We carried out additional experiments mixing non-motile *A. baylyi* with motile ones and observed interesting spatial structures similar to the flower-like patterns in mixtures of *E. coli* and motile *A. baylyi*. Therefore, it appears that indeed our proposed mechanism of pattern formation is still applicable to this situation, although the patterns in mixtures of two *A. baylyi* strains were not as pronounced and robust as in mixtures of *E. coli* and *A. baylyi*. There could be several factors responsible for this: (i) the growth rate of non-motile *A. baylyi* is less than that of *E. coli* (actually, it is even smaller than that of the motile *A. baylyi* due to burden from the selection marker), so the band of non-motile *A. baylyi* at the perimeter of the colony is not as thick as *E. coli*’s; (ii) the effective friction coefficient of non-motile *A. baylyi* may be smaller than that of *E. coli* and close to that of motile *A.baylyi*. We added a new Appendix 3 (Mixtures of motile and non-motile *A. baylyi*) to discuss these new experimental results. We also added Video 8 to show the development of patterns in a mixture of motile and non-motile *A. baylyi*.

Reviewer #2:1) It would be helpful to discuss the origin of the expansion force, since only part of the colony is motile. Are twitching cells pushing the other strain through cell-to-cell contact? (can the authors provide a high resolution image to show that cells are highly packed?) Or – given that an extracellular matrix is mentioned – is it entropically driven? (Is an extracellular matrix knock-out available to the authors?)

Both mechanisms (entropically-driven expansion and pushing each other through direct cell-cell contacts) are not mutually exclusive. Based on our experimental findings that *A. baylyi* colonies expand but *E. coli* ones don’t, despite having comparable growth rates, we believe that the core reason for the colony expansion is motility, or effective temperature, i.e. it is entropically driven. Colonies do not begin to expand until they have completely covered the interior surface in a monolayer (see new Figure 1—figure supplement 2). Therefore, we believe the entropic expansion acts through direct cell-cell contacts, when cells constantly bump and push each other. Because bacteria continuously grow and divide, the cell density remains high throughout the colony expansion. ECM has been shown to entropically drive phase separation and dispersion in non-motile, single-species colonies (Seminara et al., 2011, Dilanji et al., 2014, Ghosh et al., PNAS 2015), but this does not appear to be a driving factor here, given the dependence on both motility (Figure 3E) and a mixture of two species (Figure 1C). We have now added a discussion of the origin of the expansion force in the Discussion section.

2) Are wetting forces negligible? An order-of-magnitude estimate of the different forces or experimental evidence would justify choice of these and not other forces.

We are not sure about the importance of wetting forces, however the drastic difference between the expansion of motile *A. baylyi* colonies compared with *E. coli* colonies and non-motile mutant *A.baylyi* colonies (that presumably should have similar wetting properties), leads us to believe that passive wetting forces probably do not play an important role in the process. It leaves us with the (entropic by nature) expansion force and the resistance (bottom friction, viscosity) forces that comprise the force balance in the form of Stokes equation. We added such discussion in the Discussion section.

It is true that wetting forces could play a role in expansion of the colony onto new regions of agar. In principle, *A. baylyi* could generate more surface wetting, locally aiding expansion, and *E. coli* could generate less, or even inhibit wetting. This could indeed be part of the effective friction forces associated with the different cell types – with *A. baylyi* having less effective friction and *E. coli* having more. The expansion and friction forces are left here as largely phenomenological, though experimentally motivated. It would certainly be worthwhile to further explore the physical underpinnings of these forces, but we believe that is an area for future work, although worth including in the Discussion, which we have done in the revised version.

3) In the more refined model, 2D concentrations of the two strains and of the phase field are evolved in time. I would imagine the two strains move relative to one another (*E. coli* remains attached to the substrate whereas *A. baylyi* twitches). A two-fluid mixture appears a natural choice; the authors consider instead a single velocity field carrying all cells, and I am unclear what this velocity represents. Is there a way to relate it to actual velocities measurable experimentally, e.g. a weighted average of velocities of the two strains? Because I am confused about the meaning of v, I do not have an intuition why it follows Stokes equation.

Our close inspection of the growing colony showed that the velocities of the two strains in close proximity are very similar, which is not very surprising since the mixture of *E. coli* and *A. baylyi* is dense (new Figure 1—figure supplement 2), liquid-like, and miscible. We saw no evidence of the two strains expanding at different velocities: *E. coli* were carried with the surrounding *A. baylyi* (hitchhiking) or pushed by it in the front of the colony. Thus, we decided that it would make more sense to use a single local velocity, which represents the actual velocity of the bacterial cells. Since the single-velocity model already appears sufficient in capturing the salient features of the phenomenon, we did not see a reason to make it more complex than necessary. We mention this rationale in the revised version of Appendix 2.

4) The authors predict that at low friction, the colony expands quicker with no flower like pattern, unless a large concentration of *E. coli* is inoculated. Assuming agar concentration affects friction, these predictions are verified experimentally. A discussion is missing about the choice of ξ and β, which appears to me quite arbitrary. Also: is the value of η obtained by fitting the expansion rate a plausible value for a bacterial colony?

Indeed, the parameters of the model (friction and viscosity coefficients in particular) depend on a number of physical and biochemical properties of the system (the agar composition, the properties of the extracellular matrix, density of pili on the surface of bacterial cells, etc.). Thus, we selected the parameters based on fitting the observed colony dynamics, as we explain in more detail in Appendix 2. One of the parameters in the stationary Stokes equation (22) can be scaled out without affecting the velocity, and we chose the basal friction coefficient of pure *A. baylyi*, \xi=1. Base on the presence of sharp kinks in the developing front structure, we concluded that viscosity plays a minor role in the dynamics, so we chose the viscosity coefficient to be small. The remaining two parameters (η and β) were chosen based on fitting the average expansion rates of colonies of pure *A. baylyi* (η) and different *A. baylyi / E. coli* mixtures (β). We provided a more detailed discussion of the model parameters in revised Appendix 2.

5) Hitchhiking is an appealing evolutionary advantage. The discussion would benefit from two additional points. First: the authors imply that flower-like patterns are directly related to hitchhiking. However from Figure 3A it looks like *E. coli* hitchhikes even in round colonies, maybe less so than in flower-like colonies? Could you quantify this aspect? Second: what are the parameters that switch hitchhiking on/off in the model? Are there evidences that these parameters vary in different strains? Could genes control these parameters? This would help elucidate whether hitchhiking may be actively controlled (e.g. strains turning off motility would benefit from correspondingly turning on hitchhiking)

Hitchhiking appears to be a necessary but not sufficient condition for flower-like pattern formation. Indeed, we observed that *E. coli* also hitchhiked in round colonies (e.g. in Figure 5F third panel), however patterns did not form, presumably because *E. coli* did not exert significant friction in those conditions. In the phase-field model that would correspond to reducing parameter \β that controls *E. coli*-dependent increase in friction. For small β, patterns indeed don’t form, while hitchhiking fully persists.

Reviewer #3:In the following, I will focus on discussing the mechanical modeling part, which falls more in my area of expertise. While the interface model has appeal due to its simplicity, there are a few things that need to be better clarified if not corrected about it (see below). The phase field modeling seems to be mostly appropriate with respect to the assumptions made, but also here at least some clarifications would be good, mostly related to the field Φ (see below).More detailed comments:1) A brief discussion on possibilities of where the force F0 and the "active pressure" in Equation 22 could originate would be good (also in the initial phase where the colony is not expanding (Figure 1D)). In subsection “Pattern-forming instability originates at the colony interface” the authors suggest some kind of active pressure created by the A. Baylyi motility. However, it is not clear to me how this alone could explain an initial phase where the colony is not expanding at all (Figure 1D). The latter seems to be more consistent with a picture of nutrient consumption combined with chemotaxis (as proposed in earlier work, Discussion section paragraph two), which could also effectively create such an F0.

The origin of the expansion force F_0_ is explained in the answer to the question 1 of reviewer 2. Briefly, it is related to the motility of *A. baylyi* so when they move randomly and bump into each other, it creates pressure field that leads to the colony expansion. However, when bacterial cells are inoculated, the cell density is low (see new Figure 1—figure supplement 2) and cells are not pushing against each other. That is the reason the colonies do not expand initially (Figure 1D). From Author response image 1 which shows the time course of the colony radius (of a mixture of *E. coli* and *A. baylyi*) obtained from numerical simulations of the phase field model with low initial density of *A. baylyi*, it can be seen that initially the colony does not expand, but once the cell density becomes large (order of 1), the colony starts to expand.

2) There are problems or at least a lack of clarity related to the way the friction force F_r_ is included in both 1D models describing the interface behavior.a) The authors assume F_r_ to be proportional to the concentration of *E. coli*, c, but independent of velocity. As a consequence, if the concentration is high enough, the surface would move inward (even without surface tension F_s_=0), driven by this friction force. How realistic is such a friction force?

The reviewer is correct, in the original formulation of the interface model the friction force F_r_ was assumed to be independent of velocity. We neglected to mention that we enforced the condition that the friction force never exceeds F_0_, so the total normal force acting on the interface was always non-negative. Thus, the interface could stall but would never move backwards. We apologize for this omission.

b) The normal velocity F[κ, g] is computed directly as a difference of normal forces (Equation 8). Apparently, the authors have implicitly assumed some additional, velocity-dependent friction here (and set the friction coefficient to one). It would be good to comment on how this additional friction is motivated.

The reviewer is also correct here; in the original formulation of the interface model we assume an overdamped motion of the interface in which the normal velocity is proportional to the balance of the pushing force and the velocity-independent friction force acting on it. Indeed, this implied that there was an additional interface resistance force proportional to the interface velocity.

c) As an alternative, one could assume F_r_ itself to depend linearly on velocity, a common assumption for motile cells on a substrate, which is also used in the authors' phase-field model (Equation 22). In this case and without additional friction, through force balance (F0-F_s_-F_r_=0) the normal velocity F[κ, g] would be given by a quotient between F0-F_s_ and a friction coefficient that is a function of c.

This is an excellent suggestion, and we decided to implement it in order to make the interface model more consistent with the phase-field model where both basal and *E. coli*-dependent friction forces are proportional to the local velocity. Thus, we assumed that the normal velocity of the interface is proportional to the sum of the constant normal force F_0_, curvature-driven surface tension and interface self-repulsion forces divided by the “friction coefficient” that has a constant (basal) term and concentration-dependent term. It only required changing two lines of code, and we did not have to change any of the model parameters. As we expected, this modification of the model did not change its stability properties, and the resultant flower-like shape of the interface remained qualitatively the same. We changed the description of the model in the corresponding section of Appendix 1, and Figure 4B,C. We are grateful to the reviewer for this valuable input that made our model more reasonable.

3) Even though the phase field approach is similar to previous work, some explanations, in particular related to the field Φ would make the manuscript more self-contained.a) In subsection “Phase-field model of flower-like pattern formation” the authors state that Φ is introduced to avoid computational difficulties of dealing with the boundary. However, these difficulties and hence the necessity of Φ is nowhere explained in more detail. Why are ρ_A_ and ρ_E_ not sufficient?

We appreciate the comment of the reviewer. Please note that the evolution of the boundary of the colony is determined by the velocity field u, which, in turn, depends on ρ_A_ and ρ_E_. Thus, it is not sufficient to have only equations for ρ_A_ and ρ_E_ but need an additional equation that governs the motion of the colony. Tracking a moving boundary using traditional techniques is challenging. For this reason, we have chosen to use the phase field method to simulate our growing colony. We have now added additional text to explain this better.

Specifically, we now write in the main text:

“It is based on PDEs for the densities of *A. baylyi* ρ_A_ and *E. coli* ρ_E_, together with an equation that describes the velocity field u of the colony. This velocity field drives the motion of the boundary of the colony and is generated by a combination of stress due to cell growth and motility, viscosity, and bottom friction that is dependent local *E. coli* density. The resulting free boundary problem is solved using the phase-field method, which introduces another PDE for an auxiliary field phi that changes continuously from 1 inside the colony to 0 outside (see Appendix 2 for the detailed formulation of the model). The boundary is then automatically defined as Φ=1/2 and can thus be computed without explicit tracking techniques.”

b) Explanations on Equation 19 can be expanded on. In particular, brief comments on the second, diffusion-like term and the last term on the right-hand side could be helpful for readers.

Our phase field description follows our earlier work on cell migration (Shao et al., 2012). To clarify this, we have added additional explanation for the different terms in Equation 19. Please note that the last term in this equation is added to stabilize the interface. This term is a computational technique, which was indeed described by Biben and Misbah, 2003 and Biben et al., 2005. We have now also added these references to the text in Appendix 3.

c) The last term in Equation 19 is known to cancel the surface tension effect created by the second term (e.g. Biben and Misbah, 2003). Why do the authors explicitly remove surface tension in their phase field model while it is present in their interface models?

The required restoring force in the interface model is provided by surface tension. Without such term, from Jacobian matrix (12), the positive eigenvalue λ(k) is proportional to k and goes to infinity as k increases. This restoring force ensures that the interface does not expand indefinitely. In our phase field model, the friction force constrains the expansion, resulting in finite expansion rates. Thus, surface tension, which can in principle be added to the phase field as a separate force term, is not necessary. We added such discussion in Appendix 3.